# LIGHT-bgcArgo-1.0: Using synthetic float capabilities in E3SMv2 to assess spatio-temporal variability in ocean physics and biogeochemistry

Cara Nissen[1], Nicole S. Lovenduski[1], Mathew Maltrud[2], Alison R. Gray[3], Yohei Takano[2,4], Kristen Falcinelli[3], Jade Sauvé[3], and Katherine Smith[2]

[1]Department of Atmospheric and Oceanic Sciences and Institute of Arctic and Alpine Research, University of Colorado, Boulder, Colorado, USA
[2]Fluid Dynamics and Solid Mechanics (T-3), Los Alamos National Laboratory, Los Alamos, NM, USA
[3]School of Oceanography, University of Washington, Seattle, WA, USA
[4]Polar Oceans Team, British Antarctic Survey, Cambridge, United Kingdom

**Correspondence:** Cara Nissen (cara.nissen@colorado.edu)

**Abstract.** Since their advent over two decades ago, autonomous Argo floats have revolutionized the field of oceanography, and, more recently, the addition of biogeochemical and biological sensors to these floats has greatly improved our understanding of carbon, nutrient, and oxygen cycling in the ocean. While Argo floats offer unprecedented horizontal, vertical, and temporal coverage of the global ocean, uncertainties remain about whether Argo sampling frequency and density capture the true spatio-temporal variability of physical, biogeochemical, and biological properties. As the true distributions of, e.g., temperature or oxygen are unknown, these uncertainties remain difficult to address with Argo floats alone. Numerical models with synthetic observing systems offer one potential avenue to address these uncertainties. Here, we implement synthetic biogeochemical Argo floats into the Energy Exascale Earth System Model version 2 (E3SMv2-LIGHT-bgcArgo-1.0). Since the synthetic floats sample the model fields at model run time, the end-user defines the sampling protocol ahead of any model simulation, including the number and distribution of synthetic floats to be deployed, their sampling frequency, and the prognostic or diagnostic model fields to be sampled. Using a six-year proof-of-concept simulation, we illustrate the utility of the synthetic floats in different case studies. In particular, we quantify the impact of i) sampling density on the float-derived detection of deep-ocean change in temperature or oxygen and on float-derived estimates of phytoplankton phenology, ii) sampling frequency and sea-ice cover on float trajectory lengths and hence float-derived estimates of current velocities, and iii) short-term variability in ecosystem stressors on estimates of their seasonal variability.

## 1 Introduction

Autonomous observing systems, such as profiling floats, sample the global ocean at much higher spatial and temporal resolution than traditional ship-based hydrography (Talley et al., 2016; Wong et al., 2020; Johnson et al., 2022). The more than 2 million profiles collected by autonomous Argo floats as part of the "Core Argo" array over the past two and a half decades have revolutionized our understanding of the spatio-temporal variability of physical ocean properties in the top 2000 m of the water

column (e.g., Jayne et al., 2017; Johnson et al., 2022). Complementing the successful implementation of a global physical float array, a "Global Ocean Biogeochemistry (GO-BGC) Array" will be implemented over the upcoming years and decades as part of the "BGC Argo" array (Roemmich et al., 2019; Claustre et al., 2020; Matsumoto et al., 2022; Schofield et al., 2022). In recent years, the deployment of >250 floats carrying biogeochemical and biological sensors within the "Southern Ocean Carbon and Climate Observations and Modeling" project (SOCCOM, 2023; Sarmiento et al., 2023) has already permitted the first quantification of spatio-temporal variability in carbon, nutrients and oxygen in the Southern Ocean (Gray et al., 2018; Arteaga et al., 2020; Su et al., 2021; Prend et al., 2022a, b), promising similar scientific advances on the global scale. Similarly, the successful deployment of regional pilot "Deep Argo" arrays, with floats sampling down to 6000 m, demonstrates the possibility to expand this technology to the sampling of the whole water column (Blunden and Arndt, 2017). All three arrays are part of the international "One Argo" program (Roemmich et al., 2019). While the advent of Argo floats represents a major step forward in the global ocean observing system with many potential uses, important uncertainties remain due to, e.g., the specific spatio-temporal coverage of the resulting dataset and uncertainties in determining float positions.

Most Argo floats profile the upper ocean approximately every 10 days, drifting with ocean circulation in between profiles at a parking depth of ~1000 dbar and being localized via the Global Positioning System (GPS) when transmitting data upon surfacing (Fig. 1 and Jayne et al., 2017). As a float's position is not known while drifting at the parking depth, its exact trajectory in between profiles cannot be resolved, which directly affects float-based velocity estimates (Gille and Romero, 2003; Ollitrault and Rannou, 2013; Wang et al., 2022; Zilberman et al., 2023). Similarly, since the presence of sea ice prevents a float from surfacing to avoid sensor damage and total float loss (Klatt et al., 2007), floats can currently not be routinely localized while under sea ice, thus limiting the utility of Argo floats in high-latitude polar regions (Chamberlain et al., 2018).

Data from the Core Argo and BGC Argo floats may still not fully characterize the spatio-temporal variability in upper ocean biogeochemical properties with high certainty despite the unprecedented spatio-temporal coverage of the global ocean. As an example, the projected float density in BGC Argo (1000 floats; Bittig et al., 2019; Roemmich et al., 2019) corresponds to approximately one float per 6° x 6° grid cell. This target density is much larger than typical spatial scales of e.g., variability in phytoplankton chlorophyll (<100 km; see McKee et al., 2022) and $p$CO$_2$ (<1 km; see Eveleth et al., 2017), complicating the extrapolation from individual float-based observations to larger spatial scales. The 10-day sampling frequency is also longer than the turnover time of phytoplankton biomass of 2-6 days (Behrenfeld and Falkowski, 1997). Similarly, biogeochemical properties, such as nutrients and carbonate chemistry are known to exhibit variability on sub-kilometer spatial scales and on time scales shorter than 10 days (Gruber et al., 2021), e.g., due to the diurnal cycle (Kawai and Wada, 2007; Torres et al., 2021), tides (Droste et al., 2022), or ocean weather (Nicholson et al., 2022). Altogether, this complicates the float-based detection of any trends in phytoplankton dynamics and carbon cycling.

Until the Deep Argo float program is fully operational, the majority of the ocean volume will remain under-sampled (Jayne et al., 2017; Roemmich et al., 2019). As a result, the observation-based detection of changes in deep-ocean heat and oxygen content and in deep-ocean ventilation are still mostly based on scarce hydrographic observations (Johnson et al., 2015; Talley et al., 2016; Roemmich et al., 2019), which introduces large uncertainties when extrapolating to the global scale. All Deep Argo floats will be equipped with sensors to measure temperature and salinity, but the fraction of floats onto which an oxygen

sensor will be mounted is uncertain (King et al., 2021). While available funding will ultimately limit both the number of floats to be deployed and the number of floats including an oxygen sensor, dedicated studies assessing the impact of float density on our ability to track large-scale changes in both deep-ocean temperature and oxygen are lacking.

Overall, in the absence of knowledge on the true distribution of physical, biogeochemical, and biological properties, uncertainties stemming from sampling frequency and density as well as imprecise localization are difficult to address with float-based observations alone, as it remains unknown how representative a given float profile is for a wider area or a longer time scale (Chamberlain et al., 2023). Model simulations are one approach to address these uncertainties. In particular, numerical models with synthetic observing systems can provide a known truth for the global distribution of any physical, biogeochemical or biological tracer, so that such models can be used as a ideal testbed to address uncertainties in sampling network design. In the past, this approach has been used to assess variability in oceanic heat content (Johnson et al., 2015; Allison et al., 2019; Garry et al., 2019; Gasparin et al., 2020), salinity distributions (Gasparin et al., 2020), the global oceanic carbon sink (Gloege et al., 2021; Hauck et al., 2023), and chlorophyll concentrations (Ford, 2021; Clow et al., 2024), demonstrating the wide range of possible scientific applications. In general, synthetic observations can be extracted either offline from time-averaged model output or online during model run time. Most published studies extracted the synthetic observations offline (e.g. Gasparin et al., 2020; Gloege et al., 2021). This approach is storage-intensive, as model fields need to be stored at high temporal frequency (often at least daily) because real-world observations always represent snapshots of ocean properties rather than time-averages, leading to higher uncertainties if lower-frequency (e.g., monthly) model fields are used to extract the synthetic observations. While this offline extraction of synthetic observations offers the advantage of being easily applicable to any model with high-enough frequency output available, extracting synthetic observation online during the model run time eliminates the uncertainty associated with assessing time-averaged model output, as such synthetic observations provide the same snapshot view of the modeled ocean as real-world observing systems do of the real ocean. Yet, since this approach requires substantial modifications of the model code, only few models have such capabilities to date (Brady et al., 2021; Clow et al., 2024).

Here, we present the new synthetic biogeochemical float capabilities (LIGHT-bgcArgo-1.0) of the Energy Exascale Earth System Model version 2 (E3SMv2). These capabilities build on the Lagrangian, in Situ Global, High-Performance Particle Tracking (LIGHT) module (Wolfram et al., 2015; Brady et al., 2021). To more closely resemble real-world Argo floats, the synthetic floats sample the model fields online during model run time, which facilitates a more realistic assessment of what floats truly "see" when they sample the ocean. The number and distribution of the synthetic floats, the sampling frequency, and the sampled variables are defined by the end-user before the start of the model experiment. After describing the implementation of synthetic floats into E3SMv2 in more detail in the following section, we will present its utility for physical, biogeochemical, and biological research questions with several case studies. These case studies address critical uncertainties related to float sampling networks, i.e., quantifying the impact of i) sampling density on the float-derived detection of deep-ocean change in temperature or oxygen and on float-derived estimates of phytoplankton phenology, ii) sampling frequency and sea-ice cover on float trajectory lengths and hence float-derived estimates of current velocities, and iii) short-term variability in ecosystem stressors on estimates of seasonal variability.

**(a)   Synthetic biogeochemical floats in E3SM**

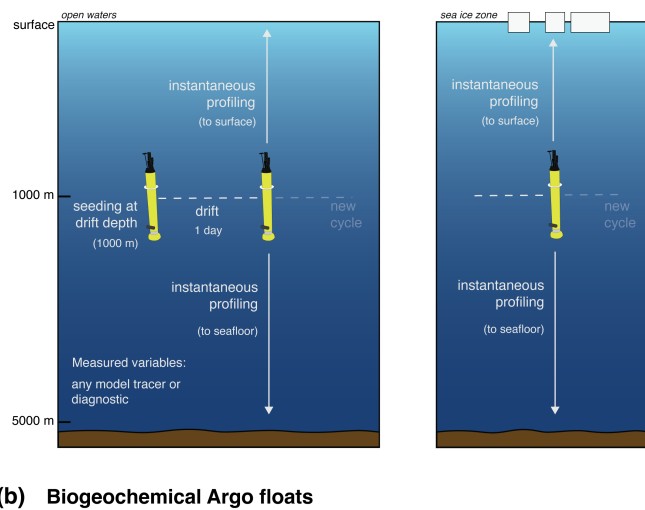

**(b)   Biogeochemical Argo floats**

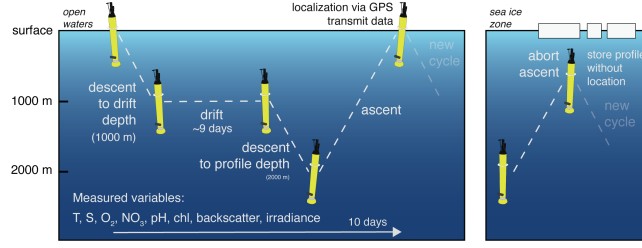

**Figure 1.** Sketch of a synthetic vs. a real-world biogeochemical float. (a) A synthetic biogeochemical float in E3SMv2 with a 1-day sampling cycle, i.e., drifting at a parking depth of 1000 m for a day and then instantaneously sampling any model tracer or diagnostic throughout the water column both in open waters and under sea ice cover. (b) A biogeochemical Argo float with a 10-day sampling cycle, i.e., drifting at a parking depth of 1000 m for about 9 days, then descending to 2000 m and sampling temperature, salinity, oxygen, nitrate, pH, chlorophyll, particulate backscatter, and irradiance in the upper 2000 m while ascending. This float relies on localization via GPS, so that its exact position is unknown while under ice. Panel (b) is adapted from Jayne et al. (2017).

## 2   Methods

### 2.1   Model description: E3SMv2 with synthetic biogeochemical floats (E3SMv2-LIGHT-bgcArgo-1.0)

We implement the synthetic biogeochemical float capabilities "LIGHT-bgcArgo-1.0" into the ocean component of the E3SMv2 version 2 (E3SMv2; Golaz et al., 2022). The physical component of E3SMv2 consists of the E3SMv2 Atmosphere Model version 2 (EAMv2), the E3SMv2 Land Model version 2 (ELMv2), the Model for Prediction Across Scales Ocean (MPAS-O), and the Model for Prediction Across Scales Sea Ice (MPAS-Seaice; Golaz et al., 2019; Petersen et al., 2019; Turner et al., 2022; Golaz et al., 2022). Both MPAS-O and MPAS-Seaice are run on unstructured multi-resolution model grids, allowing for enhanced model resolution in selected regions (Ringler et al., 2013). While we assess an ocean-sea ice-only simulation in

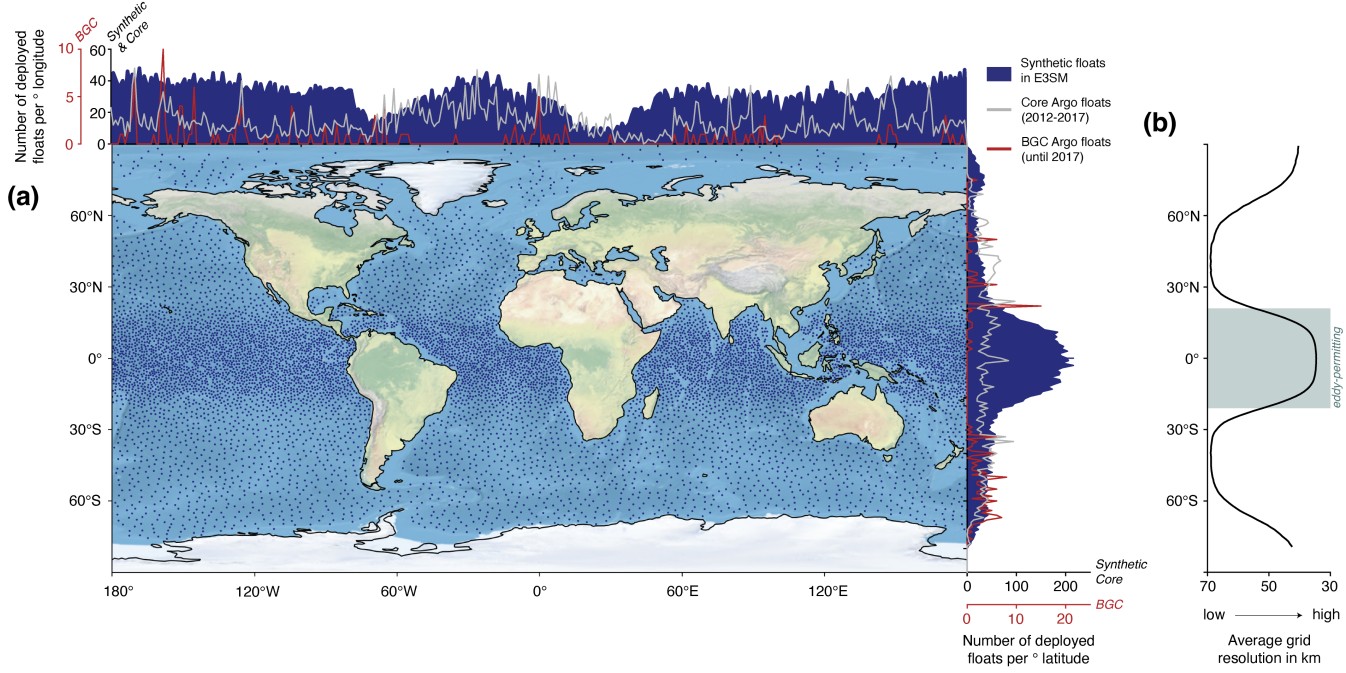

**Figure 2.** (a) Distribution of deployed synthetic floats in E3SMv2 and core and biogeochemical Argo floats between 2012-2017. Blue dots on the map indicate the initial positions of synthetic floats seeded in the deep ocean (>2000 m). On the sides of the map, the number of deployed synthetic floats per degree longitude (top) and latitude (right) is shown in blue, with the corresponding numbers for Core Argo and biogeochemical Argo floats denoted as the gray and red lines, respectively. Note the different axis scale for the biogeochemical Argo floats (red scale; synthetic and core Argo floats are shown on the black scale). (b) Zonal average model resolution in km in the E3SMv2 test simulation with synthetic floats analyzed in this paper. The shaded area denotes the latitudinal range for which the model resolution is 'eddy-permitting', i.e., where it is higher than the Rossby radius of deformation (Chelton et al., 1998).

this study, i.e., a simulation without coupled atmosphere and land model components (see section 2.2), we note that the new technical development described below can equally be used in the fully-coupled mode. Ocean biogeochemistry is described by the Marine Biogeochemistry Library (MARBL; Long et al., 2021), which is based on the Biogeochemical Elemental Cycling module (BEC; Moore et al., 2004, 2013). MARBL describes the biogeochemical cycling of carbon, nitrogen, silicon, phosphorus, iron, and oxygen and allows flexible lower trophic level ecosystem configuration (Long et al., 2021). Lastly, sea-ice biogeochemistry is represented by four dissolved inorganic nutrients (silicate, nitrate, ammonium, and iron), dissolved organic nitrogen, and two phytoplankton groups, i.e., diatoms and small flagellates (MPAS-Seaice zbgc; Jeffery et al., 2020).

The implementation of synthetic biogeochemical floats in E3SMv2 builds on the online Lagrangian, in Situ, Global, High-Performance Particle Tracking (LIGHT) module developed for MPAS-O (Wolfram et al., 2015; Brady et al., 2021). For our simulations, LIGHT particles are seeded at a depth of 1000 m and advected laterally with ocean circulation using a second order Runge-Kutta scheme; unlike previous studies (e.g., Brady et al., 2021), particles are not permitted to move vertically.

During the simulation, the virtual "floats" instantaneously sample the whole water column at their current location and with a prescribed frequency, e.g., daily or 10-day. The synthetic floats are thus not subject to lateral current displacement during ascent or descent (cf. synthetic and Argo floats in Fig. 1). Further, in contrast to Argo floats, whose position can only be registered upon surfacing in ice-free waters, the position of synthetic floats in E3SMv2 is known at all times (Fig. 1). In general, any prognostic or diagnostic physical or biogeochemical model variable can be recorded by the synthetic floats, and the sampled variables are bilinearly interpolated to the float's current location. Inclusion of profiling floats increased the computational cost of the simulation by about 50% and scaled approximately linearly with the numbers of processors, floats, and variables. However, we note that for the proof-of-concept simulation assessed in this study (see section 2.2), no attempt was made to optimize the new code's performance. In particular, interpolation weights from biogeochemical tracer locations to particle locations were unnecessarily recalculated for every tracer which certainly caused significant slowdown. The distribution of virtual particles to be seeded (or floats to be "deployed"), the sampling frequency, and the variables to be recorded are defined by the user prior to the model simulation to best align with a given application or research question.

## 2.2 Model setup and simulation

For this study, we use the ocean-ice version of E3SM, i.e., MPAS-O and MPAS-Seaice, each with their corresponding biogeochemical module (MARBL and MPAS-Seaice zbgc, see section 2.1). In this version, the ecosystem within MARBL consists of three phytoplankton functional types (diatoms, diazotrophs, and a mixed small phytoplankton group) and a single zooplankton functional type. To demonstrate the functionality of the new synthetic float tool, we conduct a six-year model simulation from 2012 to 2017, i.e., overlapping with the SOCCOM period starting in 2014 (Riser et al., 2018; Sarmiento et al., 2023). In our simulation, the model grid has a horizontal resolution that ranges from ∼30 km in the tropics and the high latitudes to ∼60 km in the subtropics (Fig. 2), and includes 60 z-star levels in the vertical, i.e., the vertical coordinate system varies with changes in the local water-column thickness in response to sea-surface height variability (EC30to60E2r2 mesh; Petersen et al., 2019). The simulation is forced with 3-hourly atmospheric data from the Japanese atmospheric reanalysis version 1.4 (JRA; Tsujino et al., 2018). All model fields are initialized from an existing, unpublished simulation using the same model grid and atmospheric forcing data as the six-year simulation analyzed in this paper (Takano et al., 2023). Model tracers in this existing simulation were initialized in the same manner as in Burrows et al. (2020). The simulation was spun-up from 1750 to 1957 using repeat-year atmospheric and river runoff forcing derived from the period July 1984 to June 1985, with atmospheric $CO_2$ concentrations held constant at 284 ppm between 1750 and 1850 and increasing according to historical records thereafter (Meinshausen et al., 2017). Starting in 1958, interannually varying JRA forcing is enabled and run through 2012, after which point the atmospheric $CO_2$ is held at a constant value of 405 ppm through 2017. Nutrient inputs with river discharge are invariant in time and taken from Mayorga et al. (2010, GNEWS model). The model uses climatological fields for atmospheric deposition of dust and iron (Luo et al., 2003) and nitrogen (Lamarque et al., 2010). Eulerian model output is stored at monthly frequency for all variables; daily output is stored for phytoplankton and zooplankton biomass.

Synthetic biogeochemical floats (a total of 10560) are "deployed" on January 1, 2012. Initially, floats are located at every grid cell vertex at a depth of 1000 m, then are successively culled so that a specified number of cells separates each float. Due to the

multi-resolution model grid of MPAS-Ocean, the resulting density of synthetic floats varies in space. Only floats seeded in the open-ocean away from continental shelves and slopes, i.e., at a water depth >2000 m, are retained for the analysis in this study

(8739 floats, see Fig. 2a). The synthetic float density is up to four times higher than the density of Core Argo floats deployed for the same period in the (sub)tropics (especially in the Pacific and Indian sectors) and comparable to it elsewhere (Fig. 2a). For all latitudes and longitudes, the synthetic float density exceeds that of Deep Argo (not shown) and BGC Argo (red lines; note the different scale; Fig. 2a). The synthetic floats instantaneously sample the whole water column every day at midnight Greenwich Mean Time. In addition to every float's position at sampling, the following model variables are recorded: temperature, salinity,

dissolved inorganic carbon (DIC), alkalinity, nitrate, silicic acid, phosphate, oxygen, total phytoplankton carbon biomass, total phytoplankton chlorophyll, zooplankton carbon biomass, $O_2$ consumption, $O_2$ production, the zonal and meridional resolved velocity components, sea-ice fraction, mixed-layer depth (as determined with the 0.03 kg m$^{-3}$ density criterion), sea-level pressure, surface partial pressure of $CO_2$ ($p$CO$_2$) in seawater, difference in $p$CO$_2$ between the ocean and the atmosphere, atmospheric $CO_2$ concentration, air-sea $CO_2$ flux, and air-sea $O_2$ flux. We calculate pH on each float profile offline using the

python routines to model the ocean carbonate system (mocsy v2.0; code was obtained from https://github.com/jamesorr/mocsy on July 19, 2023; Orr and Epitalon, 2015). We use DIC, alkalinity, potential temperature, salinity, silicic acid, phosphate, and sea-level pressure from each float profile as inputs for the mocsy functions.

## 3    Results & Discussion

### 3.1    Evaluation of synthetic float velocity, temperature, salinity and nitrate in E3SMv2

We evaluate the synthetic float capabilities in E3SMv2 in two ways: 1) by comparing the synthetic float data to the full Eulerian model output, we ensure the sampling by synthetic floats technically functions as intended and is sufficient in terms of spatio-temporal coverage, and 2) by comparing the synthetic float data to Core Argo data (Argo, 2023), we evaluate the extent to which the new synthetic observing network can be used for real-world applications. Specifically, we evaluate whether ocean currents at 1000 m, i.e., at the float parking depth, are adequately represented and whether environmental variables, such as

temperature, salinity, and nitrate, are adequately simulated in E3SMv2 for realistic float-based sampling.

The simulated pattern of current velocities in E3SM agrees with an observation-based estimate, but current speeds are overall biased low in the model. In E3SM, current velocities at 1000 m are highest in the Antarctic Circumpolar Current and in the subpolar North Atlantic off the southeast coast of Greenland (locally >8 cm s$^{-1}$), with velocities of less than 3 cm s$^{-1}$ elsewhere (Fig. 3a). Fig. 3b shows a Lagrangian-based velocity estimate derived from all 10-day synthetic float positions which were

averaged within 3°x3° boxes. The spatial patterns and magnitudes in current velocity produced by the Eulerian model output are largely captured by the synthetic floats (cf. Fig. 3a & b). The equatorial regions are the only exception, for which the Lagrangian E3SMv2 estimate suggests higher velocities (up to 4 cm s$^{-1}$) than the Eulerian estimate (up to 2 cm s$^{-1}$). This implies substantial variability of current speeds at 1000 m in E3SMv2 at sub-monthly time scales that are not captured by the Eulerian time-averaged output. Quantitatively, using bilinear interpolation to align the average Eulerian velocity field to

the same 3°x3° grid of the Lagrangian velocity estimate, the Pearson correlation coefficient between the two fields amounts

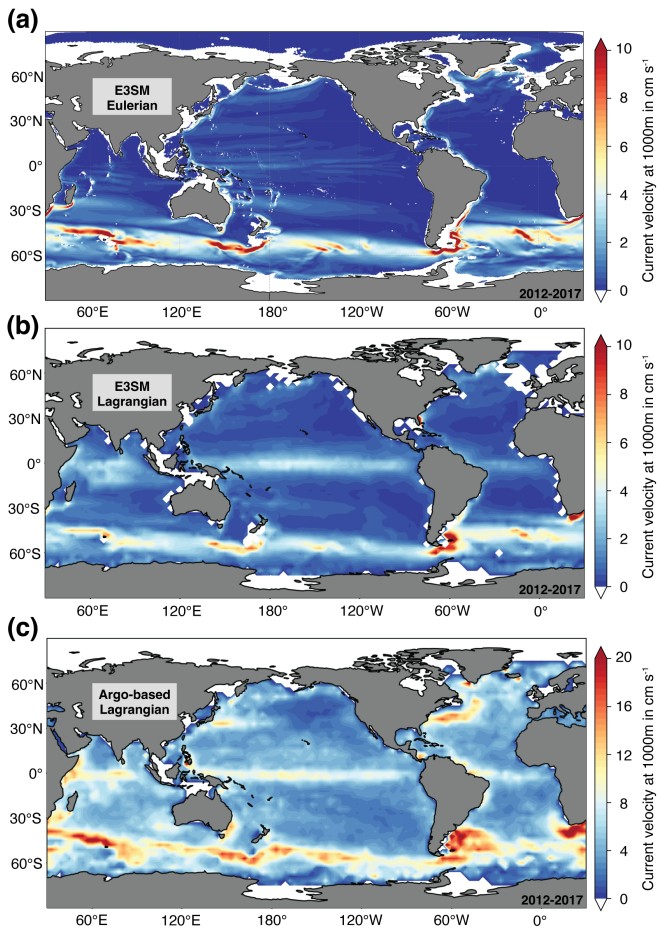

**Figure 3.** Horizontal current velocity at 1000 m in cm s$^{-1}$ (a) in the full Eulerian E3SMv2 output from 2012 to 2017, (b) derived from 10-day position data of synthetic floats in E3SMv2 and averaged for regular 3° x 3° boxes and (c) derived from 10-day position data of Argo floats and averaged for regular 3° x 3° boxes (Zilberman et al., 2022, 2023). Note the different scales between the panels (a)-(b) and (c).

to 0.40, and the area-weighted mean bias is 0.21 cm s$^{-1}$. In comparison to Argo-derived current speeds (Zilberman et al., 2022, 2023), velocities in E3SMv2 at 1000 m are a factor 2-3 too low, and the area-weighted mean bias amounts to 4.2 cm s$^{-1}$ (compare panels a-c in Fig. 3). This bias in ocean current speeds is a common feature in non-eddying ocean circulation models and is possibly related to how high-frequency dynamical processes are parameterized (e.g., internal mixing or tides; Su et al., 2023), in addition to limitations related to grid resolution. In spite of this bias, most high-velocity features present in the Argo-derived data set are reproduced in E3SMv2 (Fig. 3b & c), and the Pearson correlation coefficient between the two fields amounts to 0.66. The only exception to the fairly good spatial agreement is the Gulf Stream, which is too shallow in E3SMv2 (not shown), resulting in much lower current speeds at 1000 m in E3SMv2 than in the Argo-based estimate in the northwest Atlantic (<2 cm s$^{-1}$ compared to ~12 cm s$^{-1}$).

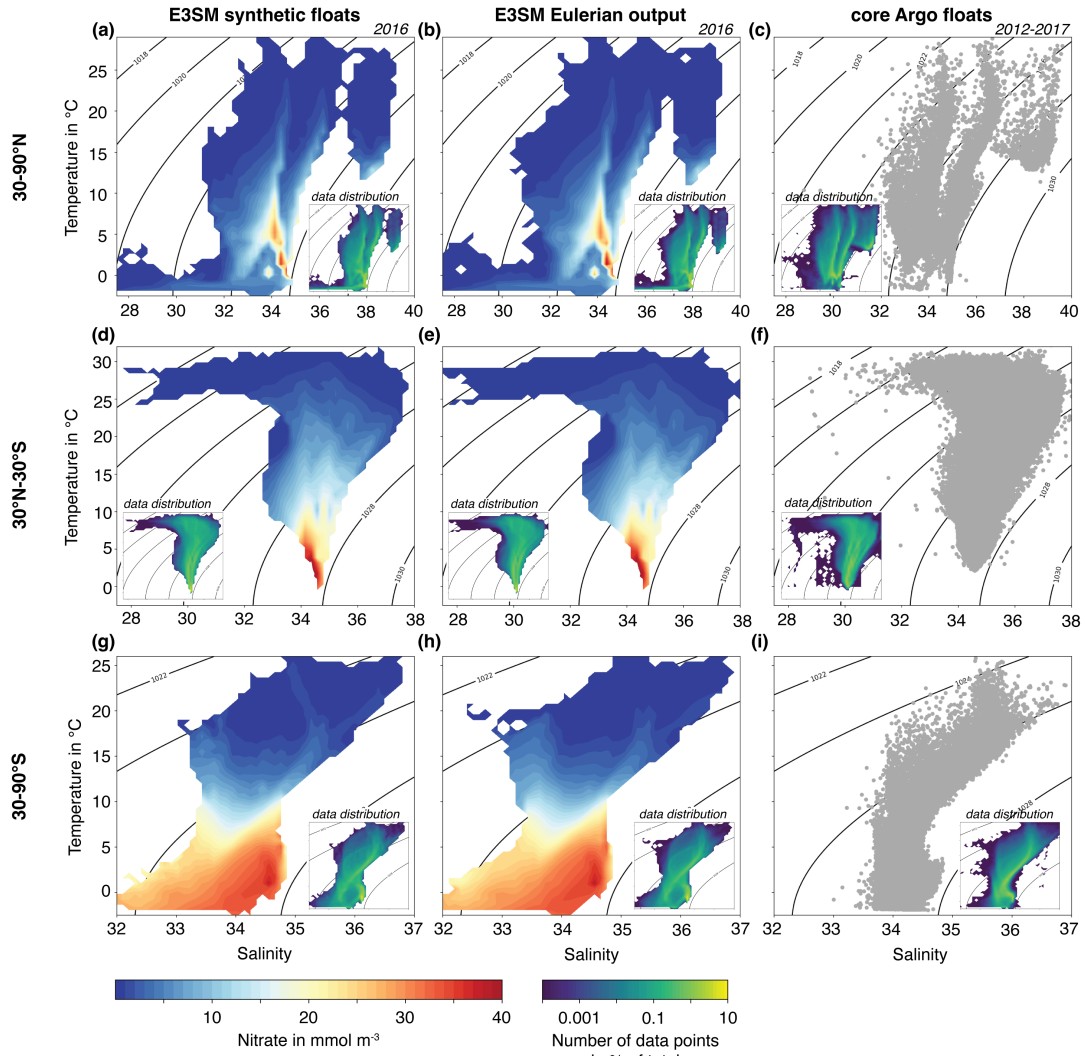

**Figure 4.** (a)-(c) Temperature-salinity-diagrams north of 30°N of (a) 10-day data from synthetic floats in E3SM, (b) monthly mean Eulerian E3SMv2 output, and (c) Core Argo float data (Argo, 2023). Data points in panels (a)-(b) are for the whole water column in 2016, and data points in panel (c) are a subset of all available points for the top 2000 m and 2012-2017. We note that subsurface interannual variability of temperature and salinity is negligible on the large spatial scale assessed here and that >80% of all E3SMv2 data are in the top 2000 m of the water column, implying comparability of the first two columns with the third. Data points in panels (a)-(b) are colored as a function of nitrate concentrations in mmol m$^{-3}$ and shown as averages within 50 equally-sized bins in temperature and salinity space. The small inlets in panels (a)-(c) show the distribution of data. (d)-(i) Same as (a)-(c) but for (d)-(f) 30°N-30°S and (g)-(i) south of 30°S.

Synthetic floats in our configuration of E3SMv2 are capable of sampling the wide range of global physical and biogeochemical properties that appear in the Eulerian-mean model output, and they sample across a wider range of global temperature and

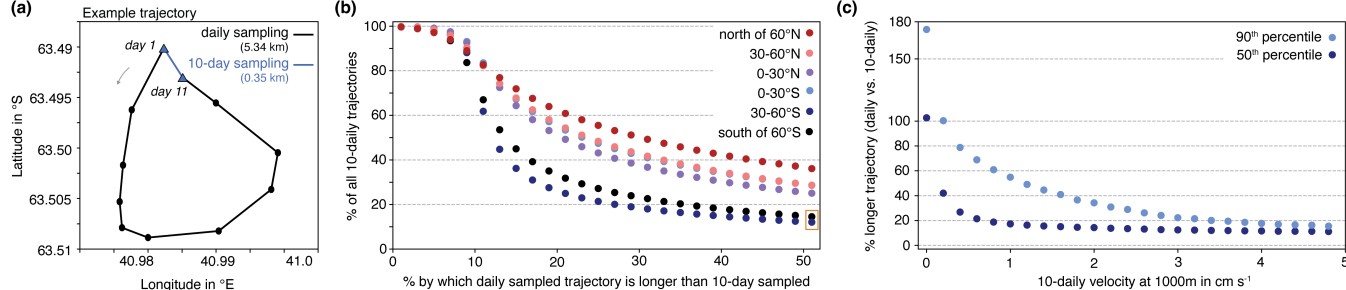

**Figure 5.** (a) Example synthetic 10-day trajectory from the Southern Ocean to compare daily and 10-day sampling. (b) Comparison of the 10-day trajectory length of all synthetic floats for daily and 10-day sampling. Plotted is the fraction of all 10-day trajectories (y axis) displaying a trajectory length at daily sampling that exceeds the length at 10-day sampling by a certain percentage (x axis). Data are grouped into 30° latitudinal bands (see colors). "% longer trajectory" as shown on the x axis is equivalent to "% lower velocity". The orange box refers to the example given in the manuscript text. (c) Difference in 10-day trajectory length of the synthetic floats for daily and 10-day sampling plotted as a function of the average 10-day velocity at 1000 m in cm s$^{-1}$ as derived from the float positions. Shown are the 50$^{th}$ (dark blue) and 90$^{th}$ percentile (light blue).

salinity values than Core Argo floats. By comparing the model data sets in temperature-salinity space, we evaluate the ability of the synthetic floats in E3SMv2 to correctly sample their model environment (Fig. 4). For the three latitudinal bands 30-90°N, 30N-30°S, and 30-90°S, the sampled temperature-salinity-nitrate space is very similar for the 10-day whole-water-column synthetic float output and the monthly mean Eulerian output (compare the first two columns in Fig. 4; note that only data for the year 2016 is shown here). We attribute any differences to not having a synthetic float sample in every single grid cell and to the differing temporal resolution of the data. In comparison to all core-Argo data 2012-2017 (third column in Fig. 4), the model output samples a larger temperature-salinity space (see e.g., cold, fresh waters for 30-90°N in panels a-c). While model biases likely contribute to some extent, we mostly attribute this difference to differences in the float distribution (e.g., in contrast to in E3SM, there are very few floats in the Arctic within Argo, see Fig. 2) and to differences in the sampled water depth (see e.g., fewer data points in Core Argo than in E3SMv2 for the latitudinal band 30-90°S at temperatures between 0°C and 5°C and salinities between 34 and 35; these data points lie in the deep ocean >2000 m as indicated by nitrate concentrations exceeding 35 mmol m$^{-3}$). In summary, the synthetic floats in E3SMv2 reproduce key large-scale patterns of variability both of the Eulerian model output and of the Core Argo floats, making these floats a valuable tool for the assessment of spatio-temporal variability in physical and biogeochemical properties from a Lagrangian perspective and for sampling network design.

### 3.2 The impact of sampling frequency on float-derived velocities

Only knowing the position of typical Argo floats upon surfacing every ∼10 days, Argo-float derived velocity estimates are subject to uncertainty stemming from the assumption of a linear trajectory between any two positions (see example from a synthetic float in Fig. 5a). With velocity calculated as the distance traveled per 10 days, a shorter trajectory length for 10-day sampling (blue line in Fig. 5a) than for daily sampling (black line) implies that the velocity derived from 10-day positions is

underestimated relative to that derived from daily positions. We use the synthetic E3SMv2 floats to compare the difference in the 10-day trajectory length (equivalent to the 10-day averaged velocity) when a) knowing the respective float's position once per day and b) only knowing the respective float's position on day 1 and day 11 (as for Core Argo floats) of each 10-day period (Fig. 5). Analysis of all our modeled synthetic floats reveals that the true distance traveled by the floats over a 10-day period is longer than indicated by their 10-day position differences. Our analysis shows that the mismatch in the trajectory length between daily and 10-day float profiling frequencies can be substantial (Fig. 5b), with 10% of all 10-day trajectories in the Southern Ocean south of 30°S being at least 50% longer when the float position is known every day as opposed to only at the start and end of the trajectory (orange box in Fig. 5b). In other words, for 10% of all Southern Ocean synthetic float trajectories, velocities derived from the float positions are more than 50% too low if the float positions are only known at the start and end of any 10-day period. In general, the smaller the error in trajectory length/velocity (x axis in Fig. 5b), the more trajectories are affected. In addition, trajectories in more northerly latitudinal bands are more severely affected by this error than those in the Southern Ocean. For example, while 80% of all trajectories north of 60°N (<70% in the Southern Ocean south of 60°S) display a difference of at least 10% in 10-day trajectory length/velocity for daily vs. 10-day float profiling frequencies, the number of affected trajectories amounts to nearly 40% (10% in the Southern Ocean) for a 50% mismatch. These regional differences reflect the general gradient from high velocities at 1000 m in the Southern Ocean (Antarctic Circumpolar Current) to lower velocities in the northern hemisphere (see Fig. 5c and Fig. 3a). Since the 10-day trajectory length forms the basis for float-derived velocity estimates (Fig. 3; Ollitrault and Rannou, 2013; Zilberman et al., 2023), our analysis illustrates the bias introduced by the absence of more frequent knowledge on every floats' position. Acknowledging that it remains unclear to what extent the absence of eddy-permitting or eddy-resolving grid resolution at extratropical latitudes affects these results (Fig. 2), our analysis demonstrates that this bias can be quite substantial in certain instances.

### 3.3 Case studies

Here, we present four example mini-studies using output from the synthetic floats in E3SMv2 to illustrate some of the capabilities of this modeling tool. In particular, we will use the synthetic floats to quantify variability in ecosystem stressors as derived from synthetic float snapshots at different sampling frequencies (section 3.3.1), the impact of float sampling density on the float-based detection of changes in deep-ocean water-mass properties (section 3.3.2), the impact of sea-ice cover on estimates of trajectory length (section 3.3.3), and the impact of float sampling density on float-derived phytoplankton bloom phenology (section 3.3.4). The analysis in each of the following subsections is not meant to be comprehensive, and many more applications are imaginable, some of which will be outlined further. Each of the following subsections will be structured like a mini-paper, with a motivation followed by methods specific to the respective case study, before presenting and discussing the results.

#### 3.3.1 Case study I: Float-based quantification of seasonal variability in marine ecosystem stressors

Our first case study quantifies the synthetic float-derived amplitude of seasonal variations of physical and biogeochemical marine ecosystem stressors, i.e., temperature, nutrient availability, oxygen levels, and carbonate chemistry, as recorded at different

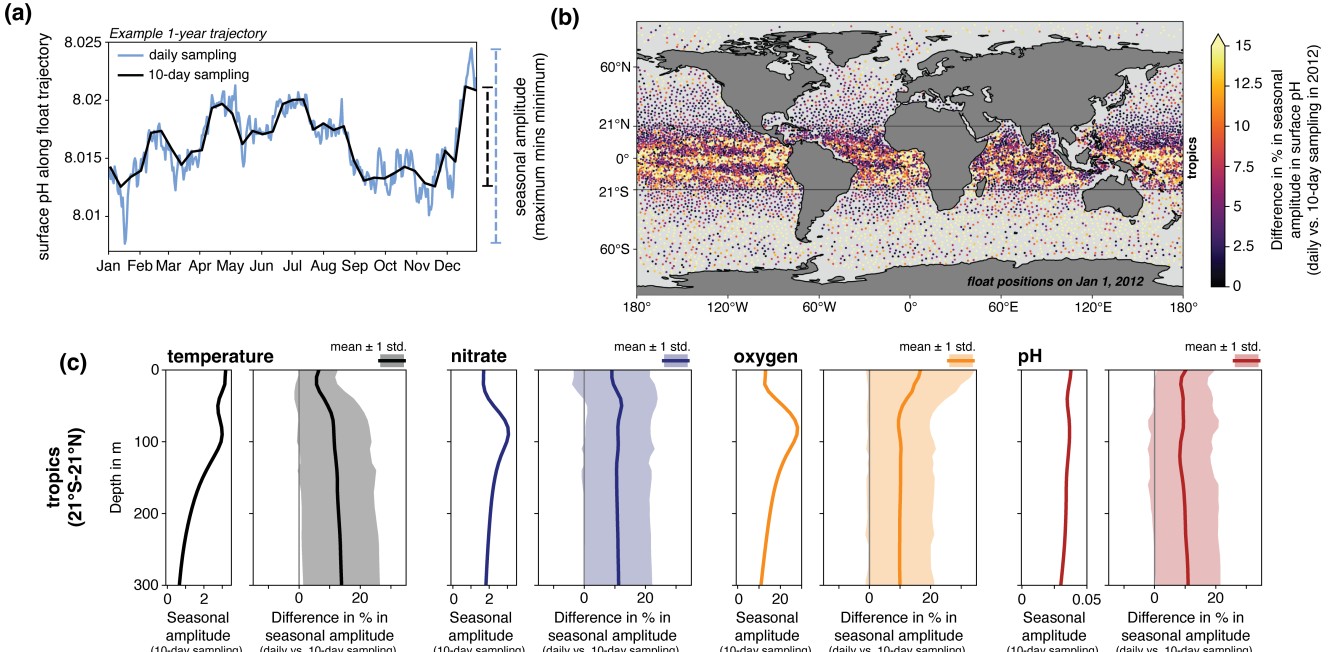

**Figure 6.** Case study "Quantifying physical-biogeochemical variability": (a) Evolution of surface pH along an example 1-year float trajectory for daily (blue) and 10-day (black) sampling. The seasonal amplitude, defined as the maximum minus the minimum pH over the whole year, is given for both sampling frequencies on the right side. (b) Map showing the difference in seasonal pH amplitude in % between daily and 10-day sampling for each float in the year 2012. (c) Synthetic float-derived vertical profiles for the top 300 m of (left) the seasonal amplitude with 10-day sampling and (right) the difference in the seasonal amplitude between daily and 10-day sampling in % for temperature (black), nitrate (blue), oxygen (orange), and pH (red). All floats which stayed in the tropical region between 21°S and 21°N for any full calendar year 2012-2017 have been included in the analysis, resulting in 34098 estimates of seasonal amplitude. The solid lines correspond to the average over all estimates, and the shading denotes one standard deviation.

sampling frequencies. This case study is motivated by the need to understand the present-day exposure of marine organisms
240   to certain environmental conditions, as this can inform their potential for acclimation and adaptation to future environmental change (Kapsenberg and Cyronak, 2019). Observations have revealed on-going change in the seasonal amplitude of upper-ocean carbonate chemistry, e.g., south of Australia (mooring-based; Shadwick et al., 2023) and in the global ocean (ship-based; Landschützer et al., 2018), making an earlier exceedance of thresholds critical to ecosystems likely and highlighting the need to better understand seasonal and subseasonal variability in all marine ecosystem stressors. Based on regional hydrographic,
245   glider, and mooring observations, we know that substantial short-term variability in ecosystem stressors is caused by, e.g., the diurnal cycle (Torres et al., 2021), tides (Droste et al., 2022), or ocean weather (Nicholson et al., 2022). High temporal resolution data can be provided by satellites for the global surface ocean (although temporal composites are often necessary due to data gaps at a sub-monthly scale; Prend et al., 2022c; Clow et al., 2024), by gliders for the upper ocean in small regions (Chai et al., 2020), and by profiling floats, which offer advances over these aforementioned technologies in terms of their spatio-

temporal coverage of the global ocean, especially at the subsurface. Yet, given the floats' 10-day sampling cycle, it remains unclear to what extent these data capture extreme conditions which are not representative of the seasonal cycle. Further, the contribution of daily variability to float-derived estimates of seasonal variability remains unquantified.

In this case study, we use the ability of the synthetic floats in E3SMv2 to sample the water column each day to assess how sampling frequency affects estimates of the seasonal amplitude of marine ecosystem stressors. In particular, we assess the difference in daily and 10-day sampling for estimates of the seasonal amplitude in temperature, nitrate, oxygen, and pH (Fig. 6). Defining the seasonal amplitude as the difference between the maximum and minimum of a given property over any given calendar year 2012-2017 (see Fig. 6a for an example), we quantify the seasonal amplitude along all one-year trajectories of synthetic floats in E3SMv2. We particularly focus our analysis on the top 300 m of the water column where seasonal variability is largest (Fig. 6c) and on the tropical region between 21°S and 21°N, where our model simulation is run at eddy-permitting resolution (see Fig. 2). Only retaining float trajectories which stay within the tropical latitudinal bounds for any full calendar year, this results in 34098 estimates of seasonal amplitude for this region, and we report the mean $\pm$ one standard deviation in Figure 6c.

The sampling frequency of the synthetic floats substantially affects estimates of seasonal amplitude, with the effect varying both horizontally (Fig. 6b) and vertically (Fig. 6c). Acknowledging that differences in seasonal surface pH amplitude of more than 15% are simulated for all ocean regions, the average difference for surface pH is largest in the tropics (Fig. 6b), where the eddy-permitting model grid resolution facilitates a stronger spatio-temporal variability in the simulated tracer fields. In the tropics, the average seasonal amplitude between 50 m and 300 m derived from daily sampling exceeds the amplitude derived from 10-day sampling by 12.4% (temperature), 11% (nitrate), 10% (oxygen), and 9.6% (pH; Fig. 6c). Above 50 m, the difference is similar to the one for below 50 m for nitrate and pH, while being smaller and larger for temperature (6.5%) and oxygen (14.7%), respectively. The larger discrepancy for oxygen is likely associated with the time scale for air-sea exchange of oxygen (a few weeks; Sarmiento and Gruber, 2006). Our findings imply that atmosphere-ocean oxygen disequilibrium manifests more strongly in seasonality estimates derived from daily float snapshots. For all marine ecosystem stressors, the variability (expressed as one standard deviation around the mean) in the impact of sampling frequency is substantial, with differences in estimates of seasonal amplitude often exceeding 20° (all variables), 25% (subsurface temperature), or even 30% (near-surface oxygen).

While it is unsurprising that higher-frequency sampling captures more temporal variability, the 10.6$\pm$10.9% larger seasonal amplitude in daily float sampling in the tropics in our model experiment (mean $\pm$ one standard deviation averaged over all ecosystem stressors in the top 300 m of the water column) highlights the uncertainty associated with the float-based detection of changes in prevalent environmental conditions in a given region based on 10-day sampling. Differences in seasonality of physical and biogeochemical properties across large-scale ocean regions are well-established (Longhurst, 1995; Fay and McKinley, 2014; Rodgers et al., 2023), but the simulated spatial variability within any large-scale region in E3SMv2 (Fig. 6b) underscores the importance of both sampling distribution and frequency when aiming to adequately capture large-scale dynamics with observations. This is especially apparent for the tropics at eddy-permitting grid resolution (Fig. 2). Yet, we note that even at non-eddy-permitting resolution, the synthetic floats suggest differences in seasonal amplitude for the two sampling

frequencies of comparable magnitude to those in the tropics for some float trajectories in, e.g., the Southern Ocean (Fig. 6b). Altogether, this suggests that while 10-day sampling with floats provides unprecedented global observational coverage to quantify spatio-temporal variability in marine ecosystem stressors, targeted regional assessments of sub-10-daily variability, e.g., with gliders (e.g., Thomalla et al., 2015) or moorings (e.g., Shadwick et al., 2023), are necessary to adequately quantify exposure of marine organisms to varying environmental conditions, which is a key factor in determining an organism's resilience to environmental change (Helmuth et al., 2014; Kapsenberg and Cyronak, 2019).

### 3.3.2 Case study II: Float-based detection of changes in deep-ocean water-mass properties

To obtain basin-scale estimates of variability and trends in the properties of climatically important water masses such as Antarctic Bottom Water or North Atlantic Deep Water, a denser observing network in both space and time is required than hydrographic observations can provide (Talley et al., 2016; Jayne et al., 2017; Roemmich et al., 2019). To that end, a key goal of Deep Argo is the detection of changes in deep-ocean heat content, which will facilitate the tracking of the deep ocean's contribution to steric sea-level rise (Johnson et al., 2015; Roemmich et al., 2019). Further, the addition of oxygen sensors on Deep Argo will enable the detection of changes in deep-ocean oxygenation, particularly in regions downstream of deep and bottom water formation in the North Atlantic and Southern Ocean (Hoppema, 2004; Rhein et al., 2017). While any difference in the spatio-temporal variability of deep-ocean temperature and deep-ocean oxygen will directly impact the number of floats required to capture large-scale changes in each variable over time, this difference remains unquantified to date.

In this case study, we use different float densities to quantify the error associated with capturing larger-scale temporal variability in deep-ocean temperature, salinity, and oxygen in the North Atlantic (between 30-60°N and 10-60°W) and Southern Ocean (south of 60°S). To facilitate the comparison of errors across variables, we calculate the normalized root mean square error (NRMSE; normalized by one standard deviation of all monthly Eulerian values averaged over the respective subarea, see Fig. 7 for spatial distribution of variables) between Eulerian and synthetic float model output. By normalizing by one standard deviation, the underlying assumption is that this metric captures sufficient variability of the true tracer distribution to facilitate drawing conclusions on the required float density to reproduce the temporal evolution of different variables. In particular, we calculate the NRMSE between the 6-year long monthly time series of the Eulerian model output and the float-derived monthly time series constructed from 10-day sampling for a given float density. For each float density, we randomly subsample all available floats in each subregion 10,000 times to obtain NRMSE percentiles of the time-series mismatch (see violin plots in Fig. 7).

Our analysis reveals that (1) the NRMSE decreases when more floats are used in the calculation of the error across all variables and in both regions, (2) the NRMSE is overall higher in the North Atlantic than in the Southern Ocean, and (3) for both regions, the NRMSE is highest for temperature and lowest for oxygen. The first point aligns with that of other studies (e.g., Youngs et al., 2023). The second point can possibly be attributed to more variability in bottom topography in the North Atlantic (note the presence of the mid-Atlantic ridge in Fig. 7a), making a float-based estimate of any water-mass property in this region more dependent on the float distribution than in the Southern Ocean, where the average spatial variability is much smaller (Fig. 7a, d, g). This finding is unaffected by the normalization of the error metric; the root mean square error (RMSE)

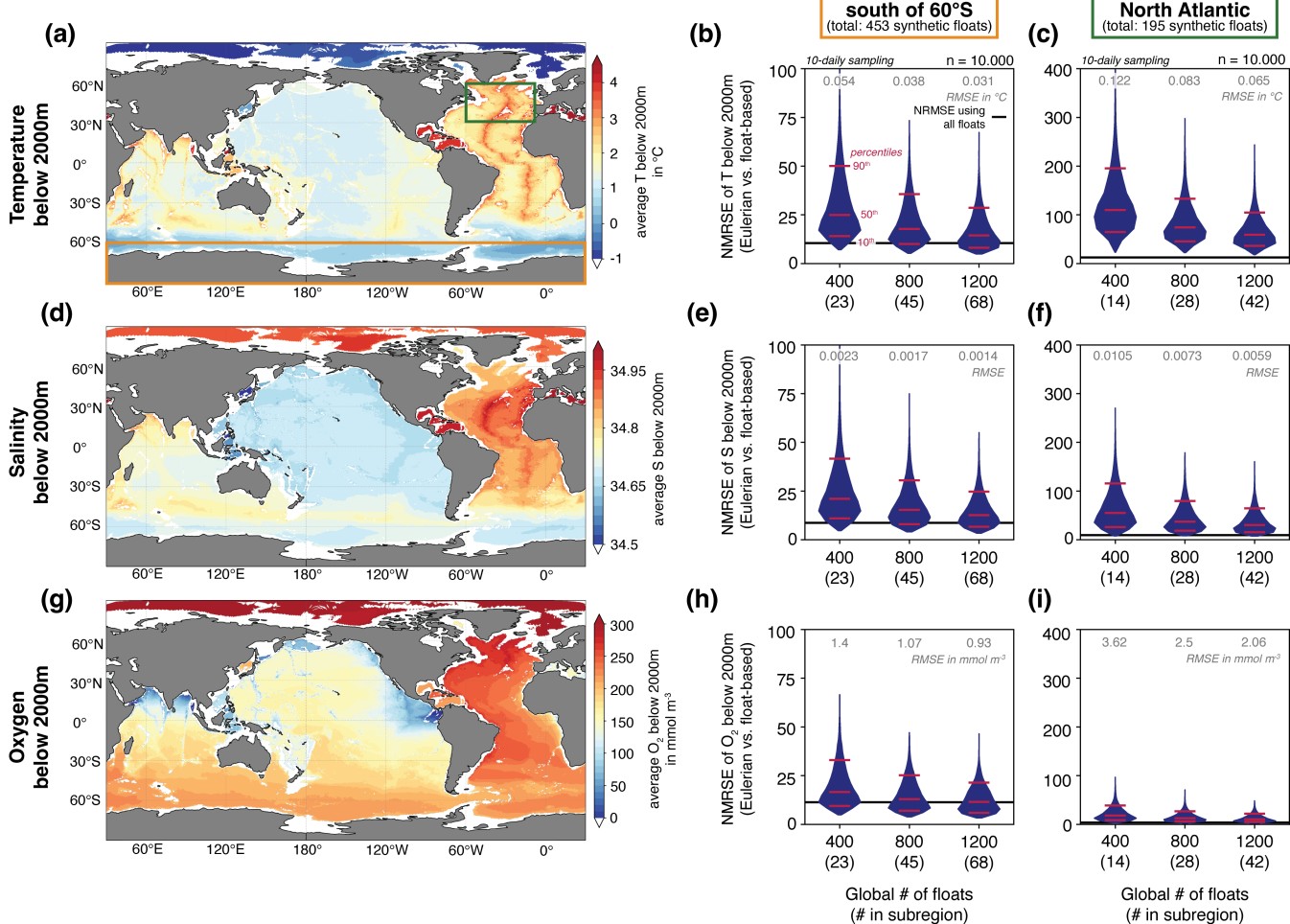

**Figure 7.** Case study "Deep ocean": (a) Temperature in °C below 2000 m and averaged over 2012-2017. (b) Violin plots of the normalized root mean square error (NRMSE) of the monthly temperature time series for the area south of 60°S (orange box in panel a) between the full Eulerian output and a float-based estimate using 10-day sampling and subsets of all synthetic floats corresponding to different global target float densities (x axis). The subsampling is repeated 10,000 times. The red horizontal lines denote the 90th, 50th, and 10th percentile (from top to bottom). The black horizontal line corresponds to the NRMSE using all synthetic floats in the area. The RMSE is normalized by the area-weighted standard deviation of monthly mean temperature values of all grid cells in the area. The unnormalized RMSE is denoted in grey. (c) Same as (b) but for the North Atlantic between 30-60°N and 10-60°W (green box in panel a). (d)-(f) and (g)-(i) Same as (a)-(c) but for (d)-(f) salinity and (g)-(i) oxygen in mmol m$^{-3}$, respectively.

is approximately two (temperature), four (salinity) and two (oxygen) times larger in the North Atlantic than in the Southern

Ocean for all float densities over the 6-year time series (see printed gray numbers in violin plots in Fig. 7).

That the NRMSE is highest for temperature and lowest for oxygen in both regions has implications for the design of a deep-ocean float array. A generally lower NRMSE for biogeochemical (oxygen) than for physical (temperature and salinity)

water-mass properties implies that, to capture property changes at comparable accuracy (at least in terms of normalized error metrics), it would be sufficient to equip a subset of all floats with oxygen sensors, thereby reducing the overall cost of a deep-

ocean float array. However, we note that based on our analysis, the RMSE for a deep-ocean array consisting of 400 floats, i.e., one third of the global target density (Johnson et al., 2015; Jayne et al., 2017), amounts to 1.4 mmol m$^{-3}$ (Southern Ocean; printed gray numbers in Fig. 7h) and 3.62 mmol m$^{-3}$ (North Atlantic; Fig. 7i), which is approximately six times larger than the range of regionally and monthly averaged Eulerian model output over the 6-year time series (0.24 mmol m$^{-3}$ and 0.60 mmol m$^{-3}$; not shown). Our analysis thus suggests that such a reduced deep-ocean oxygen array would complicate the

float-based detection of interannual variability and possibly trends in deep-ocean ventilation on large spatial scales. Importantly, the RMSE is reduced by 33% and 43% in the Southern Ocean and the North Atlantic, respectively, when equipping a full global network of 1200 floats with oxygen sensors instead of only 400 floats (Fig. 7h-i), enhancing our ability to capture variability in oxygen concentrations over large spatial scales with floats.

### 3.3.3 Case study III: Float trajectories under Southern Ocean sea-ice cover

Argo floats rely on localization via GPS upon surfacing in ice-free waters (Fig. 1b). Since the risk of damaging instrumentation is high upon contact with sea ice, conventional floats follow an ice-avoidance protocol, which makes them abort their ascent when subsurface temperatures indicate high likelihood of sea ice present at the surface (Klatt et al., 2007; André et al., 2020). In the absence of position data, the trajectory of such a float is then linearly interpolated between the last position before and the first position after the under-ice period. Depending on how long a float cannot be localized, this procedure potentially causes

large uncertainties in the estimated trajectory (Chamberlain et al., 2018; Nguyen et al., 2020).

In this case study, we use our ability to localize synthetic floats at all times, including under sea-ice cover (Fig. 1a), to quantify the impact of sea-ice presence on float trajectories for different sectors of the Southern Ocean. Using all Southern Ocean 1-year float trajectories from our 6-year proof-of-concept simulation, we compare the 1-year trajectory length of floats 1) when knowing the position of the float every day of the year and 2) when linearly interpolating a float's position when it is

under substantial sea-ice cover. For this analysis, we use the modeled sea-ice concentration as sampled by the synthetic floats to determine whether substantial sea-ice cover is present for a given location and time, and we quantify the difference in trajectory length for different sea-ice concentration thresholds defining 'substantial sea-ice cover', ranging from 5-95%. Lastly, for our analysis, we divide the Southern Ocean south of 60°S into four sectors, i.e., the Weddell Sea (between 300°E and the prime meridian), East Antarctica (0-160°E), the Ross Sea (160-210°E), and the Amundsen and Bellingshausen Sea (210-300°E).

Linear interpolation of float position data in E3SMv2 under-ice floats introduces biases in under-ice trajectory length and estimated under-ice statistical tracer properties (Fig. 8). In agreement with observations (Eayrs et al., 2019), annual mean sea-ice concentration in E3SMv2 is highest in the southwestern Weddell Sea (Fig. 8a). In winter (June-August), large areas of the high-latitude Southern Ocean are fully ice covered, implying a lack of exact position data for Core Argo floats. The trajectory of an example synthetic float in E3SMv2 reveals that the mismatch in the 1-year trajectory length can be substantial (58% in the

example in Fig. 8b, assuming unknown float positions when sea-ice cover exceeds 50%). As the linear interpolation of position data also re-locates the associated data of physical and biogeochemical water-mass properties (as illustrated with nitrate in

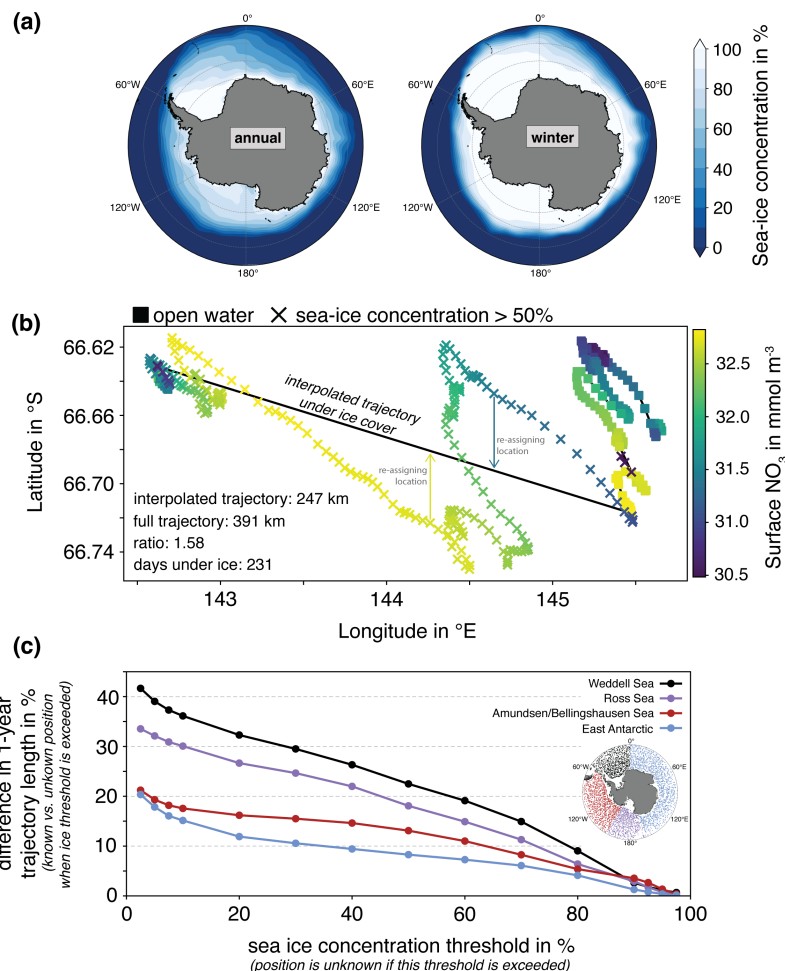

**Figure 8.** Case study "Sea-ice cover": (a) Annual mean (left) and winter mean (June, July, August; right) sea-ice concentration in percent in each grid cell of E3SMv2 averaged over 2012-2017. (b) Example illustrating the effect of unknown float positions under sea ice on the trajectory length. The daily location of the float is plotted in latitude-longitude space and colored as a function of surface nitrate concentrations (in mmol m$^{-3}$). For this example, the location is assumed unknown whenever the sea-ice concentration exceeds 50% (crosses) and known elsewhere (squares). When the float location is unknown, the trajectory is obtained by linearly interpolating between the last and first known position (black line). (c) Difference in trajectory length in % for all 1-year trajectories between 2012-2017 with known or unknown daily positions as a function of the sea-ice concentration threshold in %. The position of a float is considered unknown whenever a given sea-ice threshold is exceeded, and data gaps are linearly interpolated. Results are shown separately for the Weddell Sea (black), Ross Sea (purple), Amundsen and Bellingshausen Sea (red), and the east Antarctic (blue). See map for the initial float positions of all considered 1-year trajectories.

Fig. 8b), this procedure adds uncertainty to the float-derived tracer distribution field in the sea-ice zone, and challenges our ability to accurately determine decorrelation length scales of these tracers (as done in e.g., Eveleth et al., 2017).

The difference in under-ice trajectory length depends on the threshold used to identify critical sea-ice conditions and on the location of the float (Fig. 8c). The difference in trajectory length is larger the more often a float encounters critical sea-ice conditions. Thus, the higher average sea-ice concentration in the Weddell Sea than in other sectors explains the largest average differences in trajectory length in this region (∼40% for sea ice concentration thresholds ≤10%, see Fig. 8c). Even for a sea-ice concentration threshold of 50%, trajectories are between 10% (East Antarctic) and 25% (Weddell Sea) longer when accounting for all floats' true positions than when linearly interpolating positions under sea ice cover. Given that Core Argo floats aim to avoid any direct contact with sea ice to minimize the risk of damage, their ice-avoidance procedures are very risk-averse (Klatt et al., 2007), which corresponds to a low sea-ice concentration threshold for surfacing in Fig. 8. Acknowledging that the horizontal grid resolution used here is too coarse in the Southern Ocean to adequately resolve eddies (Fig. 2), our analysis of the synthetic floats in E3SMv2 showcases the possibly large uncertainty in float trajectories when float positions for such conditions are unknown. Alternative approaches to locating floats under sea ice, such as via acoustic tracking (Klatt et al., 2007; Chamberlain et al., 2022) or via contours of potential vorticity, sea level, or density (Chamberlain et al., 2018; Oke et al., 2022) offer the potential to reduce errors in the hydrographic measurements (Nguyen et al., 2020) and the geopositioning of the associated tracer data.

### 3.3.4 Case study IV: Deriving phytoplankton phenology from biogeochemical floats

Biogeochemical floats can provide new information about surface and subsurface phytoplankton abundance, helping to elucidate their role in global carbon and nutrient cycling. In the Southern Ocean, biogeochemical floats have already provided a more detailed description of phytoplankton phenology (including under sea-ice cover, see e.g., Arteaga et al., 2020; Hague and Vichi, 2021), phytoplankton biomass loss (Moreau et al., 2020), and net community production (e.g., Johnson et al., 2017; Su et al., 2021). Similar advances are expected in other regions (e.g., Cornec et al., 2021) as more biogeochemical floats are deployed globally (see https://www.go-bgc.org/array-status and https://maps.biogeochemical-argo.com/bgcargo/; last access June 28, 2024). As a result, we urgently need to assess our ability to capture large-scale characteristics of phytoplankton dynamics with float networks differing in density.

In this case study, we assess the ability of float networks differing in float density to capture subsurface phytoplankton bloom characteristics, i.e., the timing (day of the year) and magnitude of phytoplankton biomass peaks, in the subtropical Pacific (between 15-30°N and 120-170°W), where subsurface maxima in phytoplankton biomass are commonly observed (Cornec et al., 2021; Yasunaka et al., 2022). For each calendar year 2012-2017, we compare the timing and magnitude of maximum total phytoplankton carbon biomass at 30 m and 60 m depth between 10-daily Eulerian output and 10-daily data from synthetic floats. Synthetic float data at 10-day intervals were obtained from the daily data by randomly assigning each float a sampling start day between 1 and 10, so that floats sample the modeled ocean on different days. To obtain a statistically robust estimate of the mismatch, we subsample the 197 available synthetic floats in the subtropical Pacific 5000 times to float densities ranging from 2 to 28 in this subregion (corresponding to between 100 and 1200 floats globally). This results in 30000 estimates of bloom characteristics over the six simulation years for each float density, and we report the average mismatch $\pm$ one standard deviation in Figure 9.

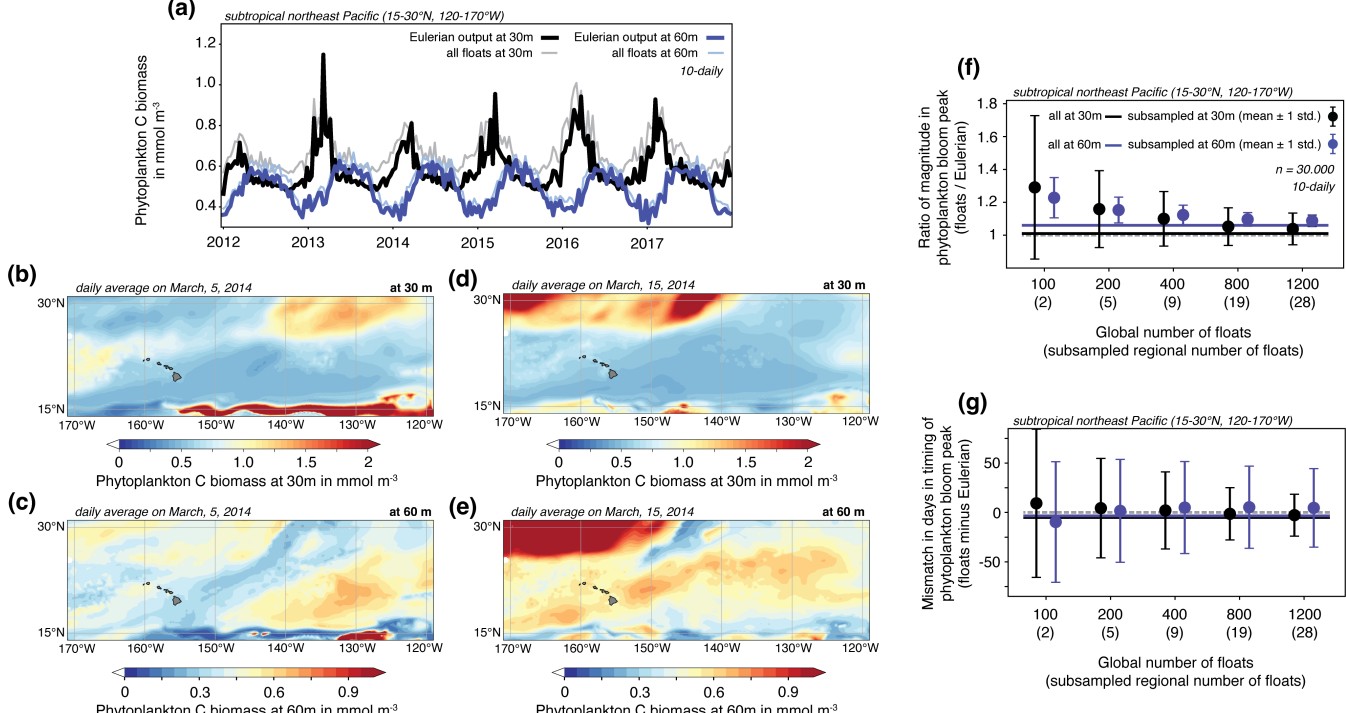

**Figure 9.** Case study "Phytoplankton phenology": (a) Time series from 2012-2017 of average 10-day total phytoplankton carbon biomass in mmol m$^{-3}$ in the subtropical northeast Pacific between 15-30°N and 120-170°W at 30 m in the full Eulerian output (black) and based on all available floats (light grey). The corresponding time series at 60 m are displayed in dark blue and light blue, respectively. (b)-(e) Example fields of daily averaged total phytoplankton carbon biomass for (b)-(c) March 5, 2014 and (d)-(e) March 15, 2014 in the subtropical northeast Pacific at (b),(d) 30 m and (c),(e) 60 m. (f) Ratio of the annual peak phytoplankton carbon biomass (mean $\pm$ one standard deviation) between the 10-day float output subsampled to different global float densities (x axis) and the full Eulerian 10-day output at 30 m (black) and at 60 m (blue) in the subtropical northeast Pacific. The subsampling is repeated 5000 times, and the ratio is thus quantified 30000 times (6 years of data) for each global float density. Horizontal lines denote the mismatch between the full float output and the Eulerian output (197 floats). (g) Same as (f), but for the mismatch in days in the timing of peak phytoplankton carbon biomass.

In the subtropical Pacific, E3SMv2 peak bloom magnitude and timing is different in the surface and subsurface ocean (Fig. 9). Peak biomass levels display more year-to-year variability and are up to two times higher at 30 m (black and grey lines in Fig. 9a) than at 60 m (dark and light blue lines). Further, based on 10-day averages, maximum biomass is simulated to occur earlier in the year at 30 m than at 60 m, possibly illustrating differences in environmental factors such as light and nutrient availability in driving phytoplankton dynamics throughout the year (Cornec et al., 2021). Overall, the spatio-temporal variability is high at both depth levels (Fig. 9a-e), suggesting that a certain number of floats is needed to adequately capture large-scale bloom dynamics in this region.

The peak bloom magnitude and timing in the subtropical northeast Pacific is dependent on the total number of floats sampling the region (Fig. 9f & g). We conduct random subsampling of the full set of 197 available 6-year synthetic float time series in the area 5,000 times for different float densities. For the vast majority of repetitions in the subsampling exercise, the magnitude of the bloom peak derived from the synthetic floats is larger than the true bloom peak derived from the Eulerian output (ratio >1 in Fig. 9f). We acknowledge that the sampling time of all floats (midnight Greenwich Mean Time) likely causes a slight systematic discrepancy between the full-day average of the Eulerian model output and the synthetic float-based estimates (see Fig. 9a). Increasing the float density improves the ability of the float network to capture the true magnitude of the bloom peak at both depths (Fig. 9f). While the mismatch at float densities of ≤200 global floats is larger for the bloom at 30 m, the opposite holds for float densities larger than this. At 60 m, even the full synthetic float network in this region, whose density corresponds to 8400 floats globally, overestimates the magnitude of the bloom peak by ∼5% (blue horizontal line), while it is well captured at 30 m (black horizontal line). On average, the timing of the bloom is reasonably well captured for all float densities (±10 days at most; Fig. 9g). However, the variability around the mean is large for both depths and all float densities (one standard deviation ranges from 21 to 75 days at 30 m and from 40 to 61 days at 60 m across float densities; see whiskers in Fig. 9g). Given that the bloom duration in the ocean typically amounts to a few weeks (see Fig. 9 and e.g., Soppa et al., 2016; Silva et al., 2021), a misrepresentation of the bloom timing of >3 weeks combined with an error in the magnitude of >10% complicate the detection of any long-term trend. In this context, we further note that the analysis of phytoplankton bloom phenology based on float-derived 10-day snapshots of phytoplankton biomass, as done here to mimic the typical real-world sampling frequency, possibly masks substantial biomass variability at sub-10-day time scales (compare panels b and d as well as panels c and e in Fig. 9). While we focus our analysis on the uncertainty in deriving phytoplankton bloom characteristics stemming from the float density in a given focus area, the synthetic float observations should also be used in future work to assess the uncertainty related to the sampling frequency. Our subtropical Pacific-focused case study highlights the spatio-temporal variability of phytoplankton dynamics in this region (McKee et al., 2022) and underlines the difficulty to adequately capture bloom dynamics on large spatial scales from sparse measurements. Future studies should investigate the magnitude of these uncertainties in other ocean regions (see e.g., Ford, 2021).

## 4 Limitations and future work

While biogoechemical float capabilities in E3SMv2 are a remarkable new tool for the ocean modeling and observational communities, the synthetic floats sample the model fields somewhat differently than Argo floats sample the real ocean. Argo floats can be laterally displaced by ocean circulation while profiling the upper ocean and while transmitting data at the ocean surface. This study does not account for the possibility of lateral displacement in the synthetic floats during this profiling, adding uncertainty to the comparison of velocity estimates derived from synthetic and Argo floats (Fig. 3). Previous work has quantified the effect of velocity shear to amount to up to 1.2 cm s$^{-1}$ in the tropics, leading to an uncertainty of ±8° in the current direction (Gille and Romero, 2003; Wang et al., 2022). Acknowledging that the absence of this effect in E3SMv2 likely has to be considered for regional applications of the synthetic floats, we assume this shortcoming to be of lesser importance for

basin-scale applications presented here. In contrast to Argo floats, all synthetic floats in E3SMv2 sample the water column at the same time each day, i.e., midnight Greenwich Mean Time. This means that the sampling of all variables undergoing strong diurnal fluctuations, e.g., light-sensitive processes such as biological productivity and hence biomass, is skewed towards a particular phase of the respective diurnal cycle, increasing the discrepancy between float-derived estimates and daily-averaged model output. As the seasonal cycle directly impacts the variability in the diurnal cycle over the course of the year, this effect is expected to be less pronounced in tropical regions (see section 3.3.4) than in polar regions.

For this study, synthetic floats in E3SMv2 are seeded uniformly as a function of grid resolution (Fig. 2). While a uniform ocean coverage by Argo float is the ultimate goal (Roemmich et al., 2019), achieving this is complicated by the dependence on ships for float deployment. In addition, in contrast to the unlimited life time of synthetic floats, the typical life time of today's Argo floats is ∼5-7 years (typically lower for BGC Argo than Core Argo; Riser et al., 2018; Roemmich et al., 2019), causing spatial gaps in our observing system if a timely re-deployment of a float is not possible for a given region. While we did not account for random failures of some sensors or entire floats in the 6-year proof-of-concept simulation analyzed for this paper, "imperfect" synthetic float data sets could easily be constructed offline to assess the impact of a temporary or permanent absence of observations in a specific region or for a specific time period. Future work should assess the impact of data scarcity and of using different mapping methods on reconstructed fields of biogeochemical tracers and, e.g., air-sea $CO_2$ fluxes (Gloege et al., 2021; Hauck et al., 2023; Heimdal et al., 2023). Lastly, we note that on smaller spatial scales, model biases and structural limitations of the E3SMv2 configuration used here due to e.g., model resolution and parametrizations of internal wave dynamics, could reduce the utility of the synthetic biogeochemical float capabilities as an ideal test bed, as the simulated variability on small spatial scales might differ from the variability experienced by real-world Argo floats.

## 5    Conclusions

We implement synthetic biogeochemical float capabilities into the Energy Exascale Earth System Model version 2 (E3SMv2-LIGHT-bgcArgo-1.0). The synthetic floats are advected with ocean circulation at 1000 m online during the model integration, sampling all desired prognostic and diagnostic model variables throughout the whole water column at the frequency prescribed by the end-user. Using E3SMv2 with synthetic floats as a perfect test bed, in which the true distribution of modeled physical, biogeochemical and biological variables is known, we demonstrated the utility of this new tool in different use cases. In particular, acknowledging the remaining uncertainty stemming from the used model configuration not explicitly resolving eddy dynamics, we demonstrated that by sampling every ten days, float-derived velocities are biased low (>50% for some 10-day trajectories). Similarly, 1-year Southern Ocean float trajectories derived from linearly interpolated float positions under sea-ice cover differ substantially from the true simulated float trajectories, especially in regions of high sea-ice cover such as the Weddell Sea ($> 40\%$ mismatch for conservative sea-ice thresholds). We further showed that on average, synthetic daily float-based snapshots of marine ecosystem stressors in the tropics result in 10.6±10.9% larger estimates of seasonal amplitude than 10-day snapshots (mean ± one standard deviation for all ecosystem stressors between the surface and 300 m). Lastly, our

results highlight the importance of the float network size for adequately capturing spatio-temporal biogeochemical dynamics, e.g. for detecting trends in deep-ocean heat content, deep-ocean ventilation, or upper-ocean phytoplankton bloom dynamics.

Even though differences exist in how synthetic and Argo floats sample the simulated and real ocean, respectively, the synthetic floats in E3SMv2 can be used in the future to improve our understanding of how Argo floats "see" the ocean, thereby contributing to the interpretation of existing observational records. For example, the synthetic float capabilities could be used

to i) assess uncertainties in deriving biogeochemical fluxes such as air-sea $CO_2$ exchange or net community production from float-based observations, ii) assess uncertainties in mapping float-based observations or derived quantities to global, gridded datasets arising from, e.g., float distributions, sampling frequency, or sensor inaccuracies including drift, or iii) inform future float deployment strategies within the One Argo program. Computing synthetic floats online eliminates both the need to store high-frequency model output and the uncertainty associated with time-averaging model output to extract synthetic observations

offline. Given that Argo floats do not sample time-averaged water-mass properties but provide a snapshot view of the ocean, the online computation produces a more realistic data set of synthetic observations. Expanding the online synthetic observing capabilities in E3SMv2 or other models by further sampling methodologies, e.g., ship-based hydrography, deep-sea moorings, gliders or surface drifters, should be a key focus of future work, with the aim to improve our global ocean observing system.

*Code and data availability.* The model source code of E3SMv2 including synthetic biogeochemical float capabilities is available at Zenodo:

Energy Exascale Earth System Model Program (2023). The synthetic biogeochemical float data set and the corresponding Eulerian model fields from E3SMv2 are deposited in the "PetaLibrary" of the University of Colorado Boulder and can be accessed via Globus (www.globus. org). To find the data, enter "E3SM-BGCArgo" as the name of the collection in the file manager.

*Author contributions.* C.N., N.S.L., M.M., and A.R.G. conceived the study. C.N. performed the analysis and wrote the manuscript. N.S.L. and A.R.G. acquired the funding. M.M. implemented the synthetic floats into E3SMv2 and performed the test simulation analyzed here. Y.T.

performed the Eulerian model simulation serving as the spin up for the test simulation with synthetic floats. All authors gave input on the case studies and commented on the manuscript.

*Competing interests.* The authors declare that they have no conflict of interest.

*Acknowledgements.* The authors are grateful for funding from the U.S. Department of Energy (DE-SC0022243). This research used resources of the National Energy Research Scientific Computing Center (NERSC), a U.S. Department of Energy Office of Science User

Facility located at Lawrence Berkeley National Laboratory, operated under Contract No. DE-AC02-05CH11231 using NERSC award BER-ERCAPm4003 and a high-performance computing cluster provided by the BER Earth System Modeling program and operated by the Laboratory Computing Resource Center at Argonne National Laboratory. Data storage supported by the University of Colorado Boulder "PetaLi-

brary". Argo data were collected and made freely available by the international Argo project and the national programs that contribute to it. SOCCOM data were collected and made freely available by the Southern Ocean Carbon and Climate Observations and Modeling (SOCCOM)

Project funded by the National Science Foundation, Division of Polar Programs (NSF PLR-1425989, with extension NSF OPP-1936222), and by the Global Ocean Biogeochemistry Array (GO-BGC) Project funded by the National Science Foundation, Division of Ocean Sciences (NSF OCE-1946578), supplemented by NASA, and by the International Argo Program and the NOAA programs that contribute to it. The Argo Program is part of the Global Ocean Observing System (https://doi.org/10.17882/42182, https://www.ocean-ops.org/board?t=argo).

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
