# Peer review of "LIGHT-bgcArgo-1.0: Using synthetic float capabilities in E3SMv2 to assess spatio-temporal variability in ocean physics and biogeochemistry"

_Geoscientific Model Development, 2023_

## Author Comment (AC1)

**Answers to comments by reviewer #1**

**General comments:**
This paper briefly describes the implementation of synthetic float capabilities in E3SMv2, then presents insights gained from the synthetic floats in several case studies. The manuscript is interesting and well written. However, even though I loathe gatekeeping based on journal scope, I do question whether the current form of the manuscript is suited for GMD as a development and technical paper. I would suggest more discussion about how the floats were implemented in the model and an evaluation of related development topics, like how much using the floats impacts the computational cost and whether it scales with the number of floats or the number of output variables saved, and also a more quantitative rather than qualitative evaluation of the model simulation accuracy.

We thank the reviewer for taking the time to provide comments on our manuscript. Inclusion of profiling floats increased the computational cost of the simulation by about 50% and scaled approximately linearly with numbers of processors, floats, and variables. However, as a proof-of-concept simulation, no attempt was made to optimize the new code's performance. In particular, interpolation weights from BGC tracer locations to particle locations were unnecessarily recalculated for every tracer which certainly caused significant slowdown. As suggested, we will provide more technical details on the new model developments including the computational cost in the revised version of the manuscript. Please also see our answers to all comments below.

**Specific comments:**

Lines 80-83: It should be clearer here that only the MPAS-O and -Seaice components are being used, and not the coupled land or atmosphere. This is stated in Section 2.2, but introducing the atmosphere and land components here gives the wrong impression that they are being used.

We have added a statement in the revised section 2.1 to make that clear:

*"While we assess an ocean-sea ice-only simulation in this study, i.e., a simulation without coupled atmosphere and land model components (see section 2.2), we note that the new technical development described below can equally be used in the fully-coupled mode."*

Section 2.2: For reproducibility and meeting interests of GMD readers, this section should provide more details about the model, particularly other components of the forcing like river runoff, atmospheric deposition, and atmospheric CO2 concentration if they are included in the forcing.

Freshwater from rivers is provided with the JRA forcing; nutrients inputs with river discharge are invariant in time and taken from Mayorga et al. (2010; GNEWS model; DOI 10.1016/j.envsoft.2010.01.007). The model uses the climatology of Luo et al. (2003; DOI: 10.1029/2003JD003483) for atmospheric deposition of dust and iron, and nitrogen deposition fields of Lamarque et al. (2010; DOI: 10.5194/acp-10-7017-2010). Atmospheric $CO_2$ is held constant at 405 ppm for the duration of the proof-of-concept simulation. We will provide this information in the revised method section.

First paragraph of Section 3.1: I think this paragraph buries the lede. It begins by discussing where 1000 m velocities are high and low in the model, and then showing that these Eulerian velocities are similar to those derived from the Lagrangian floats. Only then does it note that the model velocities are 2--3x lower than estimated from actual Argo floats. These seems to me to be a critical bias, and the impacts of it should be discussed and not glossed over.

We thank the reviewer for this remark. In the revised paper, we have revised the starting paragraph of this section to more clearly state the motivation behind the evaluation (also in response to comments by reviewer #2). Further, we have revised the text on the evaluation of current velocities to more clearly state the main take-away already in the topic sentence (instead of towards the end of the paragraph). The revised first two paragraphs of this section read:

*"We evaluate the synthetic float capabilities in E3SMv2 in two ways: 1) by comparing the synthetic float data to the full Eulerian model output, we ensure the sampling by synthetic floats technically functions as intended and is sufficient in terms of spatio-temporal coverage, and 2) by comparing the synthetic float data to Core Argo data (Argo, 2023), we evaluate the extent to which the new synthetic observing network can be used for real-world applications. Specifically, we evaluate whether ocean currents at 1000 m, i.e., at the float parking depth, are adequately represented and whether environmental variables, such as temperature, salinity, and nitrate, are adequately simulated in E3SMv2 for realistic float-based sampling.*

*The simulated pattern of current velocities in E3SM agrees with an observation-based estimate, but current speeds are overall biased low in the model. In E3SM, current velocities at 1000 m are highest in the Antarctic Circumpolar Current and in the subpolar North Atlantic off the southeast coast of Greenland (locally >8 cm s-1), with velocities of less than 3 cm s-1 elsewhere (Fig. 3a). Fig. 3b shows a Lagrangian-based velocity estimate derived from all 10-day synthetic float positions which were averaged within 3°x3° boxes. The spatial patterns and magnitudes in current velocity produced by the Eulerian model output are largely captured by the synthetic floats (cf. Fig. 3a & b). The equatorial regions are the only exception, for which the Lagrangian E3SMv2 estimate suggests higher velocities (up to 4 cm s-1) than the Eulerian estimate (up to 2 cm s-1). This implies substantial variability of current speeds at 1000 m in E3SMv2 at sub-monthly time scales that are not captured by the Eulerian time-averaged output. In comparison to Argo-derived current speeds (Zilberman et al., 2022, 2023), velocities in E3SMv2 at 1000 m are a factor 2-3 too low (compare panels a-c in Fig. 3). This bias in ocean current speeds is a common feature in non-eddying ocean circulation models and is possibly related to how high-frequency dynamical processes are parameterized (e.g. internal mixing or tides; Su et al., 2023), in addition to limitations related to grid resolution. In spite of this bias, most high-velocity features present in the Argo-derived data set are reproduced in E3SMv2 (Fig. 3b & c). The only exception is the Gulf Stream, which is too shallow in E3SMv2 (not shown), resulting in much lower current speeds at 1000 m in E3SMv2 than in the Argo-based estimate in the northwest Atlantic (<2 cm s-1) compared to ~12 cm s-1)."*

Section 3.1 in general: it would be good to have more quantitative comparisons between the model results and the Argo obs or between the model Eulerian/Lagrangian results. For Fig. 3, for example, it would be nice to know some combination of correlation, mean or mean bias, and RMSE. In addition to the speed it might also be good to know how accurate the current directions are, using something like the difference in direction.

In the revised manuscript, we will report some quantitative estimates of model-observation mismatch, e.g., by calculating the metrics the reviewer listed. While revising the manuscript, we will also explore the differences in current location. However, we note that while estimates from real-world floats are affected by lateral current shear during ascent, which will impact the diagnosed current direction (see e.g., Wang et al., 2022), our synthetic float-based estimates are not, complicating the comparison. We have thus not made a decision yet whether this is worth including in a revised manuscript.

Figure 5 and the associated text took a lot of time for me to understand, and I'm not entirely sure I still do. Maybe explicitly spelling out how the two values (daily, 10-daily) are being compared would be helpful (for example, maybe the x-axis of 5a would better be labeled as something like "% by which daily sampled trajectory is longer than 10-day sampled"). Also some clarity on how a longer trajectory means a lower velocity might help.

For the revised manuscript, we have changed the x-axis label as suggested by the reviewer. Additionally, we have added a new panel to illustrate differences in the trajectory length with one example (see new Figure 5 below). Lastly, to enhance clarity, we have added two new sentences at the beginning of the section to explain how trajectory lengths relate to velocities. The new text reads:

*"Only knowing the position of typical Argo floats upon surfacing every ~10 days, Argo-float derived velocity estimates are subject to uncertainty stemming from the assumption of a linear trajectory between any two positions (see example from a synthetic float in Fig. 5a). With velocity calculated as the distance traveled per 10 days, a shorter trajectory length for 10-day sampling (blue line in Fig. 5a) than for daily sampling (black line) implies that the velocity derived from 10-day positions is underestimated relative to that derived from daily positions."*

New Figure 5 for the revised manuscript:

[Figure]

Section 3.3.1: I don't understand why we're comparing monthly Eulerian model output with daily synthetic observations when, as summarized in the first paragraph of this section, it's more common to have daily Eulerian model output and sporadic observations.

The choice to compare monthly Eulerian model output to daily synthetic observations was dictated by what full Eulerian model output we had available when doing the initial analysis. We thank the reviewer for commenting on this inconsistency, which made us realize that this choice was probably more confusing than helpful. For the revised manuscript, we will homogenize the two data streams. We are currently re-assessing this case study using two approaches, i.e., 1) averaging the synthetic float observations to monthly means and 2) re-running the 6-year test simulation to produce daily mean Eulerian output, and will decide on one of the two for the revised manuscript.

 Section 3.3.4: Is there any nuance to how the bloom timing is defined or is it just the day of year of the peak biomass?

Indeed, we define the bloom timing as the day of the year when biomass is highest. We have revised the text to make that clearer:

*"In this case study, we assess the ability of float networks differing in float density to capture subsurface phytoplankton bloom characteristics, i.e., the timing (day of the year) and magnitude*

*of phytoplankton biomass peaks, in the subtropical Pacific, where subsurface maxima in phytoplankton biomass are commonly observed (Cornec et al., 2021; Yasunaka et al., 2022)."*

**Minor suggested edits:**

Introduction: "e.g." is used very often
In the revised introduction, we have deleted some of the "e.g." from the original text.

Line 25: "have already permitted" -> "has already permitted"
Changed as suggested.

Line 54: "uncertain" or "unknown" may be a better choice than "less clear"
We rephrased this sentence to now state "uncertain".

Lines 120--121: This could explicitly state that "away from continental shelves and slopes" means "> 2000 m", at least according to the caption of Fig 2a.
The sentence now reads:

*"Only floats seeded in the open-ocean away from continental shelves and slopes, i.e., at a water depth >2000 m, are retained for the analysis in this study […]."*

Line 193: "ascertain knowledge of" -> "know" or "understand"
The revised sentence states *"[…] by the need to understand the present-day exposure […]"*.

Line 240: "any difference in the spatiotemporal variability"; difference compared to what?
In this sentence, we are referring to the differences in spatio-temporal variability of deep-ocean temperature and deep-ocean oxygen. To clarify, we have rephrased the sentence to now read:

*"While any difference in the spatio-temporal variability of deep-ocean temperature and deep-ocean oxygen will directly impact the number of floats required to capture large-scale changes in each variable over time, this difference remains unquantified to date."*

---

## Author Comment (AC2)

**Answers to comments by reviewer #2**

In its current form, the manuscript is centered around a method for Lagrangian synthetic observation that is needed and will be welcome in the community. However, the paper needs more narrative focus and, therefore, requires substantial revision to be ready for publication. The paper's flow is non-standard in that there is never a summary of the research question and many of the described methods are saved for individual case studies. The text never mentions what these case studies will be until the results are described. This left me with a general frustration and lack of context for the work. The individual case studies also felt scattershot and vaguely incomplete (the details of which can be found in the specific comments). Is this paper trying to quantify the skill of the Lagrangian method it introduces? Is this paper trying to quantify the skill of the E3SMv2 model? How do the case studies relate to the gaps posed in the introduction? There is a good scientific story here, but the writers need to work harder to find it.

We thank Paul Chamberlain for taking the time to provide comments on our manuscript. As a "development and technical paper" submitted to GMD, the main goal of our paper is the presentation of the new synthetic float capabilities of E3SMv2. The motivation for implementing this tool was to better assess what floats 'see' when they sample the ocean. While this is our ultimate research question, we present specific applications of this overarching research question in the different case studies. We note that the chosen structure for our paper is similar to the structure in other papers published in GMD (see e.g., Muntjewerf et al., 2023; https://gmd.copernicus.org/articles/16/4581/2023/gmd-16-4581-2023.pdf).

We agree with the reviewer that each of the case studies presented in the paper could and should be studied in a lot more detail (as stated in the original text at the start of section 3.3). Aiming to find an adequate balance between the technical focus of a GMD paper (see also comments by reviewer #1) and a 'standard' research paper, we decided to present these case studies in the form of mini-papers, realizing that a more detailed assessment will need to happen as part of future work.

In response to the reviewer's comments, we have improved the presentation of the case studies in the revised manuscript, while keeping the scope of a 'development and technical paper' in GMD in mind. We have made the following changes:

In the last paragraph of the introduction, we have added an outline of what case studies will be looked at to more clearly link the info given in the introduction to what is presented in the manuscript. It now reads:

*"Here, we present the new synthetic biogeochemical float capabilities of the Energy Exascale Earth System Model version 2 (E3SMv2). To more closely resemble real-world Argo floats, the synthetic floats sample the model fields online during model run time, which facilitates a more realistic assessment of what floats truly "see" when they sample the ocean. The number and distribution of the synthetic floats, the sampling frequency, and the sampled variables are defined by the end-user before the start of the model experiment. After describing the implementation of synthetic floats into E3SMv2 in more detail in the following section, we will present its utility for physical, biogeochemical, and biological research questions with several case studies. These case studies address critical uncertainties related to float sampling networks, i.e., quantifying the impact of i) sampling density on the float-derived detection of deep-ocean change in temperature or oxygen and on float-derived estimates of phytoplankton phenology, ii) sampling frequency and sea-ice cover on float trajectory lengths and hence float-*

*derived estimates of current velocities, and iii) short-term variability in ecosystem stressors on estimates of seasonal variability."*

At the start of section 3.3, we have added an outline of the case studies to be described within this section. It reads:

*"In particular, we will use the synthetic floats to quantify variability in ecosystem stressors as derived from time-averaged Eulerian model output and snapshots from synthetic floats (section 3.3.3), the impact of float sampling density on the float-based detection of changes in deep-ocean water-mass properties (section 3.3.2), the impact of sea-ice cover on estimates of trajectory length (section 3.3.1), and the impact of float sampling density on float-derived phytoplankton bloom phenology (section 3.3.4)."*

We note that we consciously decided to describe any methodological aspects of each of these case studies in this section (instead of in the method section). Thereby, we wanted to focus the method section on the new tool, which we believe is in line with the goal of GMD papers. However, we have reworked the structure of the subsections describing each case study, so that each subsection is now structured even more like a mini-paper, i.e., including a short introductory paragraph, which is followed by paragraphs describing all relevant methodological aspects (some of which were previously only described in Figure captions), the results, and the discussion. We refer the reviewer to our answer to the comment on Figure 7 below, as well as the track-change document for a complete overview of the changes.

In the conclusion section, we have added a summary of the key findings of the use cases presented in the paper:

*"In particular, acknowledging the remaining uncertainty stemming from the used model configuration not explicitly resolving eddy dynamics, we demonstrated that by sampling every ten days, float-derived velocities are biased high (>50% for some 10-day trajectories). Similarly, 1-year Southern Ocean float trajectories derived from linearly interpolated float positions under sea-ice cover differ substantially from the true simulated float trajectories, especially in regions of high sea-ice cover such as the Weddell Sea (>40% mismatch for conservative sea-ice thresholds). We further showed that on average, synthetic float-based snapshots of environmental variability result in XX% larger estimates of seasonal amplitude than time-averaged Eulerian model output. Lastly, our results highlight the importance of the float network size for adequately capturing spatio-temporal biogeochemical dynamics, e.g. for detecting trends in deep-ocean heat content, deep-ocean ventilation, or upper-ocean phytoplankton bloom dynamics."*

Note that the "XX" will be filled once we have entirely revised this case study in response to comments by both reviewers.

**Specific Comments:**

Line 1: ,and, more recently
Changed as suggested.

Line 18: the greater than
We have rephrased this sentence to now say *"the more than 2 million profiles [...]"*.

Line 24: deployment of more than 50 floats - Please recheck. SOCCOM has almost deployed 300 floats to date.
In L. 24 of the submitted manuscript, the text states "the deployment of >250 floats carrying biogeochemical and biological sensors" – we thus believe the reviewer misread the text.

Line 35: also Gille 2003 - Statistical behavior of ALACE floats at the surface of the Southern Ocean
We thank the reviewer for this reference which we weren't aware of. Added as suggested.

Line 43: Also the BGC implementation plan
We have added a reference to Bittig et al. (2019).

Line 59: These have been estimated in chamberlain et al 2023 - Optimizing the Biogeochemical Argo Float Distribution
We added a citation to this paper in the revised text.

Line 68: Daily might be too slow to capture the long tails of wind driven flux which seem to very important.
We agree with the reviewer and have revised the sentence to now state *"often at least daily"*.

Line 71: As I understand it, this is the point of your method. I think you need to expand the reasoning behind this considerably to sharpen your argument.
We have reworked the introduction to more clearly discuss the differences between online and offline extraction of synthetic observations, highlighting the advantages of the latter:

*"[...] In general, synthetic observations can be extracted either offline from time-averaged model output or online during model run time. Most published studies extracted the synthetic observations offline (e.g. Gasparin et al.,2020; Gloege et al., 2021). This approach is storage-intensive, as model fields need to be stored at high temporal frequency (often at least daily) because real-world observations always represent snapshots of ocean properties rather than time-averages, leading to higher uncertainties if lower-frequency (e.g., monthly) model fields are used to extract the synthetic observations. While this offline extraction of synthetic observations offers the advantage of being easily applicable to any model with high-enough frequency output available, extracting synthetic observation online during the model run time eliminates the uncertainty associated with assessing time-averaged model output, as such synthetic observations provide the same snapshot view of the modeled ocean as real-world observing systems do of the real ocean Yet, since this approach requires substantial modifications of the model code, only few models have such capabilities to date (Brady et al., 2021; Clow et al., 2024)."*

Line 78: "we will present its utility for the ... research questions with several case studies" I still dont understand what the research question is, or how the method you are describing addresses those questions. You have done a good job summarizing some important gaps in our understanding of BGC modeling and BGC Argo observations. I encourage you to wrap this introduction up in a tighter bow.
We appreciate this feedback. Our broad research question for the development of this tool is to better understand what floats really see when they sample the ocean (which is then applied in several specific case studies). In the revised manuscript, we have revised this paragraph to highlight the research question more clearly. Further, in response to several other comments by the reviewer, we have reworked the introduction to more clearly highlight knowledge gaps to be

addressed in our case studies and to introduce those case studies better in the final paragraph. The final paragraph now reads:

*"Here, we present the new synthetic biogeochemical float capabilities of the Energy Exascale Earth System Model version 2 (E3SMv2). To more closely resemble real-world Argo floats, the synthetic floats sample the model fields online during model run time, which facilitates a more realistic assessment of what floats truly "see" when they sample the ocean. The number and distribution of the synthetic floats, the sampling frequency, and the sampled variables are defined by the end-user before the start of the model experiment. After describing the implementation of synthetic floats into E3SMv2 in more detail in the following section, we will present its utility for physical, biogeochemical, and biological research questions with several case studies. These case studies address critical uncertainties related to float sampling networks, i.e., quantifying the impact of i) sampling density on the float-derived detection of deep-ocean change in temperature or oxygen and on float-derived estimates of phytoplankton phenology, ii) sampling frequency and sea-ice cover on float trajectory lengths and hence float-derived estimates of current velocities, and iii) short-term variability in ecosystem stressors on estimates of seasonal variability."*

We refer the reviewer to the track-change document for a complete overview of changes in the introduction.

97: This seems like a bad thing. Why did you make this choice? What are the impacts of this choice on velocity estimates?
We agree with the reviewer that by not accounting for lateral displacement, the behavior of the synthetic floats differs from that of real-world floats. We made this choice for reasons of computational efficiency. However, we emphasize that our floats are localized at the parking depth, i.e., our synthetic float-based velocity estimates are not affected by this choice. Acknowledging the effect of lateral current shear on real-world floats, which has been estimated to amount to 1.2 cm s$^{-1}$ (Wang et al. 2022), the general mismatch between velocity estimates derived from synthetic floats and Argo floats described in section 3.1 of the paper still holds.

Figure 2: above "number of floats deployed per latitude, there are 2 scales. I do not understand what is being shown here
As stated in the Figure caption, the float density for biogeochemical Argo floats is shown on the red scale. To avoid confusion, we have added this information to the Figure and revised the Figure caption to now read:

*"Note the different axis scale for the biogeochemical Argo floats (red scale; synthetic and core Argo floats are shown on the black scale)."*

New Figure 2 for the revised manuscript:

[Figure]

Line 106: So sea ice biogeochemistry is not the same as open ocean biogeochemistry? Why did you make this choice? What are its impacts?
We thank the reviewer for spotting the reported differences between the modules for sea-ice and open-ocean biogeochemistry. In fact, there was a mistake in the submitted manuscript: *Phaeocystis* is not active in the sea-ice biogeochemistry module in our setup of the model. Given that diazotrophs are not important in the sea-ice zone (see, e.g., Luo et al., 2012, Earth System Science Data, doi: 10.5194/essd-4-47-2012), the ecosystem composition is thus consistent between sea-ice and open-ocean biogeochemistry. We have corrected the method section accordingly.

Line 110: recommend defining z-star levels for context
We have added an explanation to the revised method section:

*"[...] and includes 60 z-star levels in the vertical, i.e., the vertical coordinate system varies with changes in the local water-column thickness in response to sea-surface height variability [...]."*

Line 119: This is an unstructured grid and you are deploying at every third vertex? This needs more explanation. Is the density of seeded floats inhomogeneous? Why did you make this choice?
As can be seen from Fig. 2 of the paper and as described in section 2.2 in the paper, the density of deployed synthetic floats differs in space. We have decided to deploy the synthetic floats this way to capture high-resolution variability in regions with higher grid resolution. We have clarified this in the revised manuscript:

*"Due to the multi-resolution model grid of MPAS-Ocean, the resulting density of synthetic floats varies in space. The synthetic float density is up to four times higher […] "*

Line 135: At this point, I now know that you are going to use a BGC model with lagrangian observations, but I still dont understand what your research question is or how you intend to use these langrangian observations or why you would want to. Methods section has done a good job describing the WHAT of your analysis. Needs to be expanding to include the HOW.

We kindly refer the reviewer to our answer to the general comment above.

Line 135: I recommend being more specific in your evaluation criteria. Something like Evaluation of synthetic float temperature, salinity, and velocity.
We changed the title of the section to "Evaluation of synthetic float velocity, temperature, salinity and nitrate in E3SMv2".

Line 136: Which synthetic float data? BGC profiles should be identical to the Eulerian model, unless I am missing something very fundamental.
No, these are not identical. Since the synthetic float profiles are extracted during model run time from E3SMv2, these profiles provide snapshots of ocean properties, while the Eulerian model output provides the time-averaged view. Since the synthetic float capabilities are a new addition to E3SMv2 and given the aim of our paper as a "development and technical paper" submitted to GMD, we therefore first need to ensure that the synthetic floats are extracted correctly during model run time and that the synthetic floats are seeded at high enough density to sample the complete environmental space simulated by the model. Additionally, to determine to what extent results from the synthetic observing system hold for the real-world float observing system, we also need to establish how well the sampled environmental space by the synthetic floats agrees with real-world Argo floats. We have revised the start of the section to clarify:

*"We evaluate the synthetic float capabilities in E3SMv2 in two ways: 1) by comparing the synthetic float data to the full Eulerianmodel output, we ensure the sampling by synthetic floats technically functions as intended and is sufficient in terms of spatio-temporal coverage, and 2) by comparing the synthetic float data to Core Argo data (Argo, 2023), we evaluate the extent to which the new synthetic observing network can be used for real-world applications."*

Line 142: The case studies that you go through in this paper are never summarized in either the introduction or methods. This is problematic for 2 reasons, first I dont have context for where you are going with this analysis, second there are some details here that have never been explained. How do you calculate your lagrangian based velocities? Davis 1998 showed that there could be biases in these results if you arent careful with your distributions.
In the revised manuscript, we outline the specific case studies in the last paragraph of the introduction and at the beginning of section 3.3. Further, we summarize their outcome in the conclusions section. Please see our response to the general comment for more detail.

As stated in L. 141/142 of the submitted manuscript, we derive the Lagrangian-based velocity estimate from the 10-day synthetic float positions. We have modified this sentence to also state that float-derived velocities were then averaged within 3°x3° boxes. The sentence reads:

*"Fig. 3b shows a Lagrangian-based velocity estimate derived from all 10-day synthetic float positions which were averaged within 3°x3° boxes."*

Figure 3: The point of this plot is to intercompare velocities derived from different estimates, as such I strongly recommend changing to same colorscale.
We will optimize the color scale in the revised manuscript, although we note that it might be difficult to use the same color scale while still ensuring that patterns in both are visible.

Line 147: Recommend making units of all three colorbars identical for easier comparison
We will ensure to use the same (or more similar) spacing of the color scale in the revised manuscript.

Line 155: Assuming that E3SMv2 is perfect. This is a big assumption.
This sentence only referred to the comparison of the synthetic float data to the full model output. As such, for this assessment, there is no need for E3SM to perfectly compare to the Argo data set, which we believe the reviewer is referring to. We apologize for the confusion and have rephrased as follows to clarify:

*"By comparing the model data sets in temperature-salinity space, we evaluate the ability of the synthetic floats in E3SMv2 to correctly sample their model environment (Fig. 4)."*

Line 159: Could spatial sampling be an issue as well?
Agreed. We have adapted the sentence to

*"We attribute any differences to not having a synthetic float sample in every single grid cell and to the differing temporal resolution of the data."*

Line 161: So, you are assuming a perfect model t-s distribution? Didnt you just mention that ACC currents are wrong by a factor of 4 and the gulf stream is too shallow.
We have rephrased this sentence to now read

*"While model biases likely contribute to some extent, we mostly attribute this difference to differences in the float distribution […] and to differences in the sampled water depth […]."*

Line 167: I dont understand this conclusion or even the point of this section. Are you testing the model against Argo? Are you testing the subsampled lagrangian points against the model? I think you are missing a logical step. First, you need to validate the model against observation. Then you need to validate your subsampling against the model.
With the synthetic float capabilities of E3SM being the focus of this manuscript, we have decided to evaluate our model simulation in the Lagrangian space, i.e., by comparing the synthetic floats to Argo floats. Given that the synthetic float capabilities are a new addition to the model, the first step in this evaluation is to make sure that 1) the online extraction works and 2) the deployment strategy of the synthetic floats is adequate, i.e., enough floats are seeded to sample the full environmental space simulated by the model. Both these steps are evidenced by the good agreement between the first column in Fig. 4 and the respective panels in the second column. After this has been established (which we believe is a necessity given our manuscript is submitted as a 'technical and development paper' to a model development journal), we assess how well the sampled T-S-space by the synthetic floats compares to that sampled by available core Argo floats. Acknowledging that model biases exist on a regional scale, we conclude that the large-scale agreement is sufficient to make model subsampling exercises using the synthetic floats valuable for real-world sampling network design. We have clarified in the method section that our conclusion is drawn from the good agreement in large-scale patterns:

*"In summary, the synthetic floats in E3SMv2 reproduce key large-scale patterns of variability both of the Eulerian model output and of the Core Argo floats, making these floats a valuable tool for the assessment of spatio-temporal variability in physical and biogeochemical properties from a Lagrangian perspective and for sampling network design."*

Further, we note that we mention the short-coming of differences between the modeled and real-world small-scale variability in tracer distributions in section 4 of the manuscript. We have revised this statement to also include model biases:

*"Lastly, we note that on smaller spatial scales, model biases and structural limitations of the E3SMv2 configuration used here due to e.g., model resolution and parametrizations of internal wave dynamics, could reduce the utility of the synthetic biogeochemical float capabilities as an ideal test bed, as the simulated variability on small spatial scales might differ from the variability experienced by real-world Argo floats."*

Line 177: 50% to low?
We thank the reviewer for spotting this. Changed in the revised manuscript.

Line 180: What is the baroclinic rossby deformation radius at these latitudes? Do you think the model is resolving the eddy distribution accurately considering the resolution of your model?
Using the gridded data product by Chelton et al. (1998; Journal of Physical Oceanography) for the Rossby radius of deformation, the model grid employed here can only be considered eddy-permitting in the low latitudes. As a result, we do not think that the model is accurately resolving the eddy distribution at subpolar and polar latitudes. We acknowledge this short-coming in the revised section 3.2:

*"Since the 10-day trajectory length forms the basis for float-derived velocity estimates (Fig. 3; Ollitrault and Rannou, 2013; Zilberman et al., 2023), our analysis illustrates the bias introduced by the absence of more frequent knowledge on every floats' position. Acknowledging that it remains unclear to what extent the absence of eddy-permitting or eddy-resolving grid resolution at extratropical latitudes affects these results (Fig. 2), our analysis demonstrates that this bias can be quite substantial in certain instances."*

We have also added information on the latitudes at which the model is eddy-resolving to panel b of the revised Figure 2 (see above).

Line 185: Another bias that needs mention is the at times 70 km grid cell resolution you are using to calculate these results. Based on the substantial low-bias in figure 3, I suspect that your velocity field is far too smooth.
We agree with the reviewer. Please see our answer to the comment above, where we outline the changes made to this section of the paper.

Line 190: Its not the readers job to imagine more applications. It is the writers job to tease the scientific story out of the results.
We kindly refer the reviewer to the scope of a GMD paper and our answer to the general comments above. The main goal of our manuscript is the introduction of the new synthetic float capabilities in E3SMv2, which sample the model world online, i.e., at model run time. After the introduction of the new modeling tool, the purpose of our case studies is to illustrate with a few examples how this new tool can be used, but we believe a reader might come up with very different applications, which is why we believe such a sentence is appropriate in this case.

Figure 6: In general the text and figures are too small for me to understand what the point of this figure is
We are currently revisiting this case study in response to comments by both reviewers, i.e., we are sharpening the motivation and homogenizing the two data streams in terms of their temporal frequency. We will take this comment into account when revising Figure 6.

Figure 6: *Maps showing the ratio of the seasonal amplitude of nitrate between the daily snapshots from the synthetic floats and the monthly mean Eulerian output at those model grid*

*cells which are sampled by a float each day over any full calendar year between 2012 and 2017. Please rephrase. This does not make sense to me.*
We will rephrase the Figure captions once we have refinished revisiting this case study (see comment above).

*Line 200: , and by profiling floats,*
Changed as suggested.

*Line 201:* floats offer an advantage in temporal coverage over gliders?
We have rephrased this part of the sentence to

*"[…] and by profiling floats, which offer advances over these aforementioned technologies in terms of their spatio-temporal coverage of the global ocean, especially at the subsurface."*

Line 205: Figure 6
Changed as suggested.

Line 215: What is the sensitivity of this estimate to float distribution and sampling frequency.
We will make sure to address this comment when revisiting this case study. In particular, in response to a comment by reviewer #1, we are going to homogenize the temporal frequency of both data streams.

Line 217: Absolute variability is smallest in the deep. Perhaps it would be illuminating to explain why you are using this metric?
Acknowledging that temporal variability (on all scales) is larger in the upper ocean than in the deep ocean, it is a fair assumption that organisms *at all depth levels* are adjusted to the variability they are routinely exposed to. That means that even though a 20% difference in estimates of the seasonal amplitude translates into very different absolute changes in the deep and in the surface ocean, organisms across depth levels might be equally affected by such a change. When revising this case study, we will improve the description of its motivation and of the implications of differences in variability across depth levels.

Line 221: What exactly are you testing and why? Yes, temporal averaging smooths out high frequency variability. Why did you need to use Lagrangian particles for this result?
We agree with the reviewer that it is an expected result that temporal averaging results in less high-frequency variability, although we think that impacts of temporal averaging on estimates of seasonal variability over large spatial scales have not often been assessed quantitatively. With higher-frequency temporal variability historically having been difficult to observe (especially at the subsurface), floats offer a unique opportunity to start quantifying this variability on larger spatial scales. This is essential to ultimately better constrain the variability in ecosystem stressors that organisms are routinely exposed to, which can be an indicator for their ability to adapt to environmental change. With this case study, we aim to quantify the difference in seasonal variability captured from float-based measurements and full Eulerian model output.

Line 225: This should go in the introduction as it frames your work. Dont save it for the end.
We have included this information in the revised introduction. The sentence reads:

*"Similarly, biogeochemical properties, such as nutrients and carbonate chemistry are known to exhibit variability on sub-kilometer spatial scales and on time scales shorter than 10 days*

*(Gruber et al., 2021), e.g., due to the diurnal cycle (Kawai and Wada, 2007; Torres et al., 2021), tides (Droste et al., 2022), or ocean weather (Nicholson et al., 2022)."*

Line 231: Why not compare 10 day float BGC sampling to 1 day float BGC sampling?
We will consider this comment when revisiting this case study.

Figure 7: What is a subregion?
As stated in the caption of Fig. 7, the two analyzed subregions in this analysis, i.e., the Southern Ocean and the North Atlantic, are depicted by the orange and green boxes, respectively. To enhance clarity, we have included the definition of the two subareas in the revised text:

*"In this case study, we use different float densities to quantify the error associated with capturing larger-scale temporal variability in deep-ocean temperature, salinity, and oxygen in the North Atlantic (between 30-60°N and 10-60°W) and Southern Ocean (south of 60°S)."*

Figure 7: recommend moving description of method used for calculation to methods
As outlined in our answer to the general comment, we have decided to keep methodological aspects that only concern the case studies in section 3.3. Yet, to enhance clarity, we have double-checked the description of each case study to ensure that all information on the approach and methods used in each one is described in a "methods" paragraph for each case study. This new structure is also outlined in the revised introductory text in section 3.3. The new sentence reads:

*"Each of the following subsections will be structured like a mini-paper, with a motivation followed by methods specific to the respective case study, before presenting and discussing the results."*

As an example, for section 3.3.2, the "methods" paragraph reads:

*"In this case study, we use different float densities to quantify the error associated with capturing larger-scale temporal variability in deep-ocean temperature, salinity, and oxygen in the North Atlantic (between 30-60°N and 10-60°W) and Southern Ocean (south of 60°S). To facilitate the comparison of errors across variables, we calculate the normalized root mean square error (NRMSE; normalized by one standard deviation of all monthly Eulerian values averaged over the respective subarea, see Fig. 7 for spatial distribution of variables) between Eulerian and synthetic float model output. In particular, we calculate the NRMSE between the 6-year long monthly time series of the Eulerian model output and the float-derived monthly time series constructed from 10-day sampling for a given float density. For each float density, we randomly subsample all available floats in each subregion 10,000 times to obtain NRMSE percentiles of the time-series mismatch (see violin plots in Fig. 7)."*

We refer the reviewer to the track-change document for a complete overview of changes made to the other subsections.

Line 248: These are not metrics that I am familiar with and I do not understand the specifics of the calculation. I recommend putting a description of this analysis in the methods section and expanding it.
As described in response to the previous comment, we have included a paragraph (or expanded it in case it already existed) for each case study describing the used methods in more detail.

Line 259: I believe there is a hidden assumption in using NRMSE as a metric to quantify the number of floats required to reproduce the modeled field and that is the covariances are known perfectly. Typically, they are not. I think this should be mentioned.
To obtain the NRMSE, we normalize the RMSE between the Eulerian and the float-based estimate by the standard deviation of all monthly Eulerian values averaged over the respective subarea. The underlying assumption is thus that this standard deviation captures sufficient variability of the underlying true field. We have added a sentence to section 3.3.2 to clarify this:

*"[...] By normalizing by one standard deviation, the underlying assumption is that this metric captures sufficient variability of the true tracer distribution to facilitate drawing conclusions on the required float density to reproduce the temporal evolution of different variables."*

Line 267: The analysis presented so far is about resolving seasonal means, not trends. I agree that more sensors will always help, but I think it is important to clearly make this distinction.
In this case study, we assess to what extent monthly means as obtained from 10-day synthetic float profiles over the six-year model simulation match up with the time series of monthly mean Eulerian model output. While we acknowledge that this time series is too short to look at longer-term trends, we do include an assessment of our ability to reconstruct interannual variability. We have revised the sentence to clarify:

*"Our analysis thus suggests that such a reduced deep-ocean oxygen array would complicate the float-based detection of interannual variability and possibly trends in deep-ocean ventilation on large spatial scales."*

Line 285: how does this compare to wong and riser 2011?
We thank the reviewer for pointing us to this reference. Unfortunately, we cannot compare our estimates of the trajectory mismatch between knowing float positions under ice and linearly interpolating under-ice positions to the results in Wong & Riser (2011), as the floats assessed in their study were not localized under sea ice. Wong & Riser report that the 19 floats assessed in East Antarctica typically had to store up to 40 profiles while under sea ice. At a 7-day sampling cycle, this corresponds to 280 days under sea ice, which agrees reasonably well with the 231 days under ice shown for the example in Fig. 8b of our manuscript.

Line 301: With a greater than 1/3 degree model, you are likely missing a lot of eddy activity in the high latitude. How might that impact your results?
We agree with the reviewer on this point. Unfortunately, without having output from a higher-resolution model available to us at the moment, it is impossible for us to know what impact resolving eddies would have on our results. We have rephrased this part of the paper to explicitly acknowledge this uncertainty:

*"Acknowledging that the horizontal grid resolution used here is too coarse in the Southern Ocean to adequately resolve eddies (Fig. 2), our analysis of the synthetic floats in E3SMv2 showcases the possibly large uncertainty in float trajectories when float positions for such conditions are unknown."*

FIgure 9: Again, I recommend moving method descriptions from figure caption to methods section.
In the revised manuscript, we have included a dedicated methods paragraph in each subsection of section 3.3. Please see the track-change document and our answers to comments above.

Line 333: were synthetic float observations staggered? or did they all sample on the same day every 10 days?
The synthetic float observations were staggered. We clarify this in the revised manuscript by adding the following sentence to section 3.3.4:

*"Synthetic float data at 10-day intervals were obtained from the daily data by randomly assigning each float a sampling start day between 1 and 10, so that floats sample the modeled ocean on different days."*

Line 339: Recommend looking at Ford 2021
We added a citation to this reference.

Line 347: also Gille 2003

Added as suggested.

Line 365: The model has a 70km grid at the largest. I think you are parameterizing a lot more than internal wave dynamics.
The sentence was rephrased to

*"Lastly, we note that on smaller spatial scales, model biases and structural limitations of the E3SMv2 configuration used here due to e.g., model resolution and parametrizations of internal wave dynamics, could reduce the utility of the synthetic biogeochemical float capabilities as an ideal test bed, as the simulated variability on small spatial scales might differ from the variability experienced by real-world Argo floats."*

Line 381: This distinction became more clear to me throughout the paper, but should be highlighted in the introduction.
We kindly refer the reviewer to our answer to the comments above and to the revised manuscript. We have reworked the text to clarify the advantage of using online float capabilities to construct synthetic float datasets.

---

## Author Response (AR2)

**Answer to reviewer 1**

The revisions have improved the manuscript and I am satisfied with the responses to my previous review. I've noted a few technical details and edits that should be checked before final publication.

We thank the reviewer for the positive assessment of our revised manuscript. Please find our answers to all detailed comments below. We note that in addition to both reviewers' comments and upon suggestion by the editor, we have made some further minor edits to the title, the abstract and the main text so that the revised manuscript now includes a title for the new synthetic float capabilities ("LIGHT-bgcArgo"). Please see the track change document for an overview of these text modifications.

Line 75: insert period between "ocean" and "Yet".
Inserted as suggested.

Line 205: "only knowing the respective float's position on day 1 and 10 [...] of each 10-day period"; technically, if samples were taken exactly every 10 days, we would know the position on day 1 and day 11.

We thank the reviewer for pointing this out, which made us realize that there was a mistake in how we calculated the trajectory length from daily float positions. In the submitted manuscript, we had calculated the trajectory length between day 1 and 10, while it should of course be between day 1 and 11. We note that we had correctly calculated the trajectory from 10-daily sampling, and Figure 3 (trajectory-based velocity estimates) is thus not affected by this mistake. In the revised manuscript, we have corrected the text in section 3.2 to read "[…] on day 1 and 11 […]" and have updated Figure 5 accordingly. Further, while we have made minor additional adjustments to the text in section 3.2 to reflect the corrected calculation (see track-change document), the major take-away message of this section remains unchanged.

Section 3.3.1: Only a suggestion: this section calls the range between the maximum and minimum value during a year the "seasonal amplitude". I would call it the "seasonal range".

We thank the reviewer for this suggestion. We decided to stick to "seasonal amplitude" but have revised the starting sentence of section 3.3.1 to now read:

*"Our first case study quantifies the synthetic float-derived amplitude of seasonal variations of physical and biogeochemical marine ecosystem stressors […]"*

Line 341: "flaots" -> "floats"
Corrected.

Lines 384--385: I was confused what "float densities ranging from 2 to 28" means. Clarifying that this is the number of floats sampled from within the basin (I think) might help.

We have revised this sentence to now read:

*"To obtain a statistically robust estimate of the mismatch, we subsample the 197 available synthetic floats in the subtropical Pacific 5000 times to float densities ranging from 2 to 28 in this subregion (corresponding to between 100 and 1200 floats globally)."*

Figure 9a: the float estimates appear to have a higher mean value than the Eulerian values. Is this from a sampling bias in the floats?

Thanks for spotting this. We believe that this is at least in part due to the fact that all our synthetic floats currently sample at the same time of the day (midnight GMT), causing a slight

systematic discrepancy between the full-day average of the Eulerian model output and the synthetic float-based estimates.

Line 452: "by sampling every ten days, float-derived velocities are biased high"; this should say biased low.
Corrected.

**Answer to reviewer 2**

In the present manuscript, the authors use the Energy Exascale Earth System Model version 2 to retrieve vertical synthetic profiles of various physical and biogeochemical properties in order to assess uncertainties of the spatiotemporal retrieved by the current mission of the OneArgo program in the global ocean. After evaluating the method on 3D fields of physical parameters, velocity, and nitrate, authors present four case studies in which the synthetic profiles informs on potential uncertainties that current and/or future array of real floats encounter due to their sampling scheme and mission configuration.

Manuscript was really well presented and written, and I appreciated the effort put by the authors regarding the quality of redaction and the clarity of the Figures. I feel that this study provides an innovative way and tool to assess uncertainties for studies based on Argo floats measurements, and by extension, by any autonomous platforms. In my opinion, the manuscript is suitable for publication after minor revisions (cf detailed comments below).
We thank the reviewer for the positive assessment of our revised manuscript. Please find our answers to all detailed comments below. We note that in addition to both reviewers' comments and upon suggestion by the editor, we have made some further minor edits to the title, the abstract and the main text so that the revised manuscript now includes a title for the new synthetic float capabilities ("LIGHT-bgcArgo"). Please see the track change document for an overview of these text modifications.

l.15: It is not clear to me what the authors imply by "seasonal variability"? Do they mean: short-term variability in ecosystem stressors impact on the estimates of their seasonal cycle?
Yes, this is what we mean. To clarify, we have corrected this as suggested.

l.19 and 22: Maybe mention before and/or after presenting the 3 missions that they are part of the international OneArgo program (Roemmich et al., 2019), and mention "Core Argo" mission or array (l.19), and "BGC Argo" mission or array (l.22) for more clarity.
We have made the suggested changes and added the following sentence to the first paragraph of the introduction:

*"All three arrays are part of the international "One Argo" program (Roemmich et al., 2019)."*

Figure 1: Add "irradiance" (one of the six "core" variable of the BGC mission) on the list of parameters on the panel b.
Added as suggested.

Figure 2: I feel that it could be informative for the reader to see the distribution of the current fleet (BGC/core/Deep) on the map, in order to compare it with the synthetic floats distribution. I know that there is already a lot of information on this Figure, so this is up to the authors.
We appreciate this suggestion. We agree that it could be a nice addition to also show the Argo float distributions on the map, but we believe this would make the map too busy. The panels at

the top and on the right side of the map were included in this figure to specifically facilitate the comparison of the float distributions in Argo and in E3SM, and we hope this level of detail is sufficient for the majority of readers.

l.226-229: I would reo-order the presentation of the case study exemples so that they match the order in which they are described afterward in the manuscript.
We thank the reviewer for spotting this. While the order of the case studies did already correspond to the subsequent subsections, the references to the subsections given in the parentheses were incorrect. We fixed this in the revised manuscript.

Case 1: I agree with the fact that a 10-day sampling captures less variability over than a sub-10-days sampling scheme, but I would also suggest adding that the 10-days sampling might capture some "extreme" events that for example does not represent the average seasonal cycle, which can be important depending on the application.
We thank the reviewer for raising this point. In the revised manuscript, we have revised the first paragraph of section 3.3.1 to now read:

*"Yet, given the floats' 10-day sampling cycle, it remains unclear to what extent these data capture extreme conditions which are not representative of the seasonal cycle. Further, the contribution of daily variability to float-derived estimates of seasonal variability remains unquantified."*

l.333: I would not specify a temperature threshold, to remain "global" (the mentioned one is mostly used for the Southern Ocean, but other are used for other area of the global ocean, cf https://archimer.ifremer.fr/doc/00658/77029/).
We thank the reviewer for pointing this out. As suggested, we have taken out the mentioning of a specific temperature threshold. Further, we have added the suggested reference (André et al., 2020, DOI: 10.3389/fmars.2020.577446).

l.341: typo (end of the line): synthetic floats (instead of flaots).
Corrected.

Figure 8: I am not familiar with the sea ice concentration, but are there some units for the color scale?
Sea ice concentration represents the fraction of the respective grid cell that is covered by sea ice. As such, it can be presented as a non-dimensional property on a scale from 0 to 1 or be given in percent. To clarify, we show sea-ice concentration in percent in the revised Figure 8 of the manuscript and have modified the figure caption accordingly:

*"Annual mean (left) and winter mean (June, July, August; right) sea-ice concentration in percent in each grid cell of E3SMv2 averaged over 2012-2017."*

l. 375: I would mention the international BGC-Argo program/mission/array and not exclusively the US contribution (through GO-BGC), as other countries contribute to the program worldwide… Here is a link for a map with the implementation of the international BGC program https://maps.biogeochemical-argo.com/bgcargo/.
We have added the suggested link. The sentence now reads:

*"Similar advances are expected in other regions (e.g., Cornec et al., 2021) as more biogeochemical floats are deployed globally (see https://www.go-bgc.org/array-status and https://maps.biogeochemical-argo.com/bgcargo/; last access June 28, 2024)."*

In the different cases study, it is unclear why the authors mention 1200 floats, when the goal of the OneArgo program regarding the BGC floats operational fleet size is 1000 floats. Could the authors explicit this choice?

While the reviewer is of course correct that 1000 is the target number for BGC floats, the target number for Deep Argo is 1200 (see, e.g., https://argo.ucsd.edu/oneargo/). Acknowledging that it is still unclear how many of these floats should (ideally) be equipped with biogeochemical sensors, we have decided to report our results for a float density up to 1200 floats globally to encompass both Argo programs. Additionally, we note that for our last case study, substantial uncertainty in capturing phytoplankton bloom phenology remains even at a global BGC float density of 1200.

Fig. 9: Typo in the y axis of panels f and g: phytoplankton bloom peak (instead of phtoplankton)
Corrected.

Conclusion: I feel that the authors should emphasize more the utility of the modeled synthetic profiles for further applications using Argo data. The authors remind the results of the examples of applications, but I feel that they should generalize more the conclusion and focus rather on the applications potential of this approach, that could be use to help constraining uncertainties in future Argo studies (e.g., to constrain lateral advection of water masses in 1-D framework BGC approaches, gas exchanges estimations, etc…), as well as maybe be a potential tool to identify locations of interest for ongoing and future float deployments in the framework of the OneArgo program.

We have added the following sentence to the conclusion section:

*"For example, the synthetic float capabilities could be used to i) assess uncertainties in deriving biogeochemical fluxes such as air-sea $CO_2$ exchange or net community production from float-based observations, ii) assess uncertainties in mapping float-based observations or derived quantities to global, gridded datasets arising from, e.g., float distributions, sampling frequency, or sensor inaccuracies including drift, or iii) inform future float deployment strategies as part of the One Argo program."*

---

## Author Response (AR3)

**Answer to comments by editor**

Thank you for revising your manuscript title and model name. However, per our guidance, can you please add a version number as well? Given that this model will likely be revised into the future, it is important that the version described here has a unique identification. Perhaps "LIGHT-bgcArgo-1.0" or similar?

We have added the version number "1.0" as suggested throughout the manuscript.

Referee 1 noted: "Figure 9a: the float estimates appear to have a higher mean value than the Eulerian values. Is this from a sampling bias in the floats?". In your response you provided a satisfactory answer to this point, but this information - which may assist other readers - does not appear in the manuscript (or, at least, I cannot find it). Could you please add this information?

We have added the following sentence to the result section 3.3.4:

*"We acknowledge that the sampling time of all floats (midnight Greenwich Mean Time) likely causes a slight systematic discrepancy between the full-day average of the Eulerian model output and the synthetic float-based estimates (see Fig. 9a)."*